# Constitutive activity of an atypical chemokine receptor revealed by inverse agonistic nanobodies

Claudia V. Perez Almeria [1], Omolade Otun[1,2], Roman Schlimgen [3], Thomas D. Lamme [1], Lotte Di Niro [1], Caitrin Crudden [1], Jan Paul Bebelman[1], Noureldine Youssef [4], Lejla Musli[1], Shawn Jenjak[3], Vladimir Bobkov [1], Julia Drube [4], Carsten Hoffmann [4], Brian F. Volkman [3], Sébastien Granier [2], Cherine Bechara [2,5], Marco Siderius [1], Raimond Heukers [1,6,7], Christopher T. Schafer [1,7] & Martine J. Smit [1,7] ✉

Stimulation of atypical chemokine receptor 3 (ACKR3) by chemokines does not activate G proteins but recruits arrestin. It is a chemokine scavenger that indirectly influences responses by restricting the availability of CXCL12, an agonist shared with the canonical receptor CXCR4. ACKR3 is upregulated in numerous disorders. Due to limited insights in chemokine-activated ACKR3 signaling, it is unclear how ACKR3 contributes to pathological phenotypes. One explanation may be that constitutive activity of ACKR3 drives non-canonical signaling through a basal receptor state. Here we characterize the constitutive responses of ACKR3 using inverse agonistic nanobodies to suppress its basal activity. These tools promote an inactive receptor conformation which decreased arrestin engagement and inhibited constitutive internalization. Basal non-chemotactic, cancer cell motility was also suppressed, suggesting a role for ACKR3 in this process. The basal receptor activity in pathophysiology may provide an alternate therapeutic approach for targeting ACKR3.

Atypical Chemokine Receptor 3 (ACKR3, formerly CXCR7) is a β-arrestin-biased chemokine receptor[1] that lacks detectable G protein activation in most cell types (with the exception of primary rodent astrocytes and human glioma cells)[1–3]. Activation of the receptor leads to phosphorylation of C-terminal serine and threonine residues by GPCR kinases (GRKs)[4–7]. These modifications are critical for coordinating arrestin coupling[8]. ACKR3 is best described as a scavenger, where its primary function is to regulate the extracellular concentrations of ligands and restrict the availability for canonical receptor activation. The receptor shares chemokine ligands with both CXCR4

(CXCL12) and CXCR3 (CXCL11), both of which drive cell migration along chemokine gradients. Scavenging by ACKR3 therefore indirectly supports chemotaxis by generating directional information and preventing overstimulation and desensitization of CXCR4 or CXCR3. This regulatory activity is dependent on GRK phosphorylation, but not arrestin engagement[4,6], suggesting the receptor might better be regarded as a GRK-biased receptor[5]. Besides chemokines, ACKR3 is activated by opioid peptides (BAM22, enkephalins, and dynorphins)[9,10] and pro-adrenomedullin derivatives (adrenomedullin and PAMP-12)[11,12], although the agonist properties of adrenomedullin has recently

[1]Amsterdam Institute for Molecular and Life Sciences (AIMMS), Department of Chemistry & Pharmaceutical Sciences, Division of Medicinal Chemistry, Faculty of Science, Vrije Universiteit, Amsterdam, the Netherlands. [2]Institut de Génomique Fonctionnelle (IGF), University of Montpellier, CNRS, INSERM, Montpellier, France. [3]Department of Biochemistry, Medical College of Wisconsin, Milwaukee, WI, USA. [4]Institute for Molecular Cell Biology, CMB – Center for Molecular Biomedicine, University Hospital Jena, Friedrich-Schiller-University Jena, Jena, Germany. [5]Institut Universitaire de France, Paris, France. [6]QVQ Holding BV, Utrecht, the Netherlands. [7]These authors contributed equally: Raimond Heukers, Christopher T. Schafer, Martine J. Smit. ✉e-mail: mj.smit@vu.nl

been disputed[13]. The wide range of natural ligands binding ACKR3 suggests a flexible binding pocket and a promiscuous receptor[14]. ACKR3 is involved in many physiological functions[15] as well as in a plethora of pathophysiological processes, including inflammatory[16], cardiovascular[17], autoimmune[18], and neurodegenerative diseases[19] in addition to different types of cancer[20]. While ACKR3 provides a cardioprotective role, its overexpression is associated with neurodegeneration in the central nervous system and poor cancer prognosis.

GPCRs play an important role in the signal transduction controlling various cancer hallmarks[21]. The CXCL12-CXCR4-ACKR3 axis plays a key role in cancer cell migration, survival, and proliferation[22,23]. Enhanced ACKR3 expression in numerous cancer types (e.g., glioma, lung, breast, colorectal, lymphoma), has been associated with the shaping of CXCL12 gradients, by internalizing with the chemokine and recycling the receptor back to the plasma membrane[24]. By these means ACKR3 appears to have a pivotal role in tumorigenesis, angiogenesis, cell adhesion, and tumor growth[25–29]. Despite ACKR3's evident role in cancer development, the specific downstream signaling pathways modulated by this receptor are still unclear. Numerous studies have suggested that CXCL12-stimulated ACKR3 signals via β-arrestin-dependent pathways activating ERK and AKT[1,30–32]. However, recent reports indicate that these may be ascribed to background CXCR4 signaling via G proteins[5,33].

In addition to chemokine-induced responses, ACKR3 displays considerable constitutive activity in the apo (empty) receptor state. The receptor readily interacts with arrestin without stimulation both in cells[14] and in vitro[34]. Without a ligand bound, ACKR3 flexibly interconverts between active and inactive conformation, which leads to basal phosphorylation by GRKs that coordinates arrestin binding[35]. Additionally, the receptor constitutively internalizes by mechanisms independent of C-terminal phosphorylation[4,5]. This internalization contributes to scavenging, but is unable to dynamically respond to large fluxes in chemokine concentration[36]. This high level of constitutive activity may explain difficulties in antagonizing the receptor, as only a handful of inhibitors have been described[34,35,37–40]. It is unknown whether the constitutive activity of ACKR3 contributes to other cellular processes and if these deviate from chemokine-induced responses. Different signaling states for constitutive and agonist-stimulated activation have been observed for the virally-encoded chemokine receptor, US28[41]. Uncharacterized signaling by the apo-receptor may play an unappreciated role in ACKR3 physiology and pathophysiology.

Here we present nanobody-based inhibitors to suppress basal activation of the atypical receptor to resolve the constitutive mechanisms of ACKR3 function. Nanobodies, also known as single domain antibodies or VHH, are the variable domains from heavy chain-only antibodies found in the Camelidae family. Nanobodies display high affinity and specificity for their target and tend to interact with non-linear, 3-dimensional epitopes[42,43]. These features make them ideal molecules for targeting and stabilizing GPCRs in specific conformational states[44–48], which may be particularly important for a promiscuous protein like ACKR3. Using advanced structural dynamics methods, we showed that the nanobodies stabilize inactive receptor conformations that correlate with inhibited basal engagement with arrestins and constitutive internalization. Inhibition of receptor constitutive activity results in slower cell motility. These data highlight potential consequences of ACKR3 basal activity.

## Results

### Basal ACKR3 engagement with arrestins is suppressed by inverse agonistic nanobodies

An antagonistic nanobody targeting ACKR3, VUN701, was recently described[49]. Here, we present two additional nanobodies, VUN700 and VUN702, which were not previously characterized. All three ACKR3 nanobodies bind the receptor extracellularly, compete with CXCL12,

and do not show cross reactivity with CXCR4 (Supplementary Fig. 1, Supplementary Table 1). Due to the bulky and relatively large binding interface of nanobodies and chemokines, nanobodies sterically prevent co-binding. As a consequence, nanobodies binding to extracellular domains of chemokine receptors generally act as antagonists[45,50–52], though some are agonists[53] or have been engineered to activate receptors[54]. To resolve the pharmacological effects of these molecules, ACKR3 engagement with arrestin was tracked by BRET between the receptor with a C-terminal nanoluciferase (ACKR3-Nluc) and β-arrestin2 C-terminally tagged with mVenus (β-arr2-mV), following addition of CXCL12 agonist or the nanobodies. Activation by the agonist CXCL12 led to a robust increase in BRET ratio, indicating a recruitment of arrestin to the receptor in HEK293T cells (Fig. 1A, B). When the cells are treated with neutral antagonist VUN701, no change in association of ACKR3 with β-arrestin was detected, consistent with its previous pharmacological classification[49]. Interestingly, VUN700 and VUN702 both decreased the BRET ratio between ACKR3 and β-arrestin2 below the measured basal interaction. This decrease in ACKR3-arrestin BRET was also observed in CHO cells, confirming the effects of VUN700 and VUN702 are not limited to the HEK293 model system (Supplementary Fig. 2). This suggests that these nanobodies are acting as inverse agonists and suppressing the previously reported constitutive ACKR3 activity[14,35]. Similar results were observed for β-arrestin1 recruitment (Supplementary Fig. 3).

ACKR3 requires phosphorylation by GRKs to engage β-arrestins in response to CXCL12[5,6,55], while a constitutively-active arrestin can interact with unmodified apo-ACKR3 in vitro[34]. To ascertain the role of GRK phosphorylation in the basal association of arrestin with ACKR3, arrestin recruitment was also tested in CRISPR-knockout cells of the four ubiquitously expressed GRKs, GRK2, 3, 5, and 6 (GRK2/3/5/6 KO)[56]. In these cells, the response to CXCL12 was completely abolished, consistent with previous results (Fig. 1C, D). Additionally, the inverse agonistic effects of VUN700 and VUN702 was no longer detectable in the absence of GRKs. Together, these data suggest that the basal arrestin association to ACKR3 is GRK-dependent and likely reflects the phosphorylation of constitutively active receptors by these kinases.

To further resolve differences between agonist-induced and basal arrestin engagement with ACKR3, the conformational changes within the arrestins were monitored using Nluc/FlAsH arrestin intramolecular BRET biosensors[57,58]. These sensors report subtle differences in arrestin conformations, corresponding to the active conformations arrestin adopts due to its interaction with GPCRs (Fig. 1E). As reported by the FlAsH 5 (F5) sensor, activation by CXCL12 promoted a robust decrease in BRET, indicating the adoption of an active arrestin state (Fig. 1F). None of the nanobodies produced a change in the signal from this sensor. Similar responses were observed for two other arrestin conformational sensors with different FlAsH positions (Supplementary Fig. 4). This implies that the basal interaction of ACKR3 and β-arrestin2 does not induce a conformational change in the arrestins and suggests the constitutively active conformation of ACKR3 is different than that promoted by agonist stimulation. Similarly distinct active conformations were observed for the viral GPCR US28[41].

### Different ACKR3 conformational states are stabilized by antagonist and inverse agonist nanobodies

The inhibition of basal arrestin interactions with ACKR3 by VUN700 and VUN702 suggests that these molecules act as inverse agonists, possibly by inducing a more inactive-like receptor conformation. Two structural dynamics methods were used to determine how the nanobodies were specifically altering the conformation of the receptor; hydrogen-deuterium exchange mass spectrometry (HDX-MS) and nuclear magnetic resonance (NMR). First, HDX-MS was performed to track changes in the rate of isotopic exchange between amide hydrogens on ACKR3 and deuterium in the solvent. The exchange rate depends on solvent accessibility and hydrogen bonding networks, and comparing these

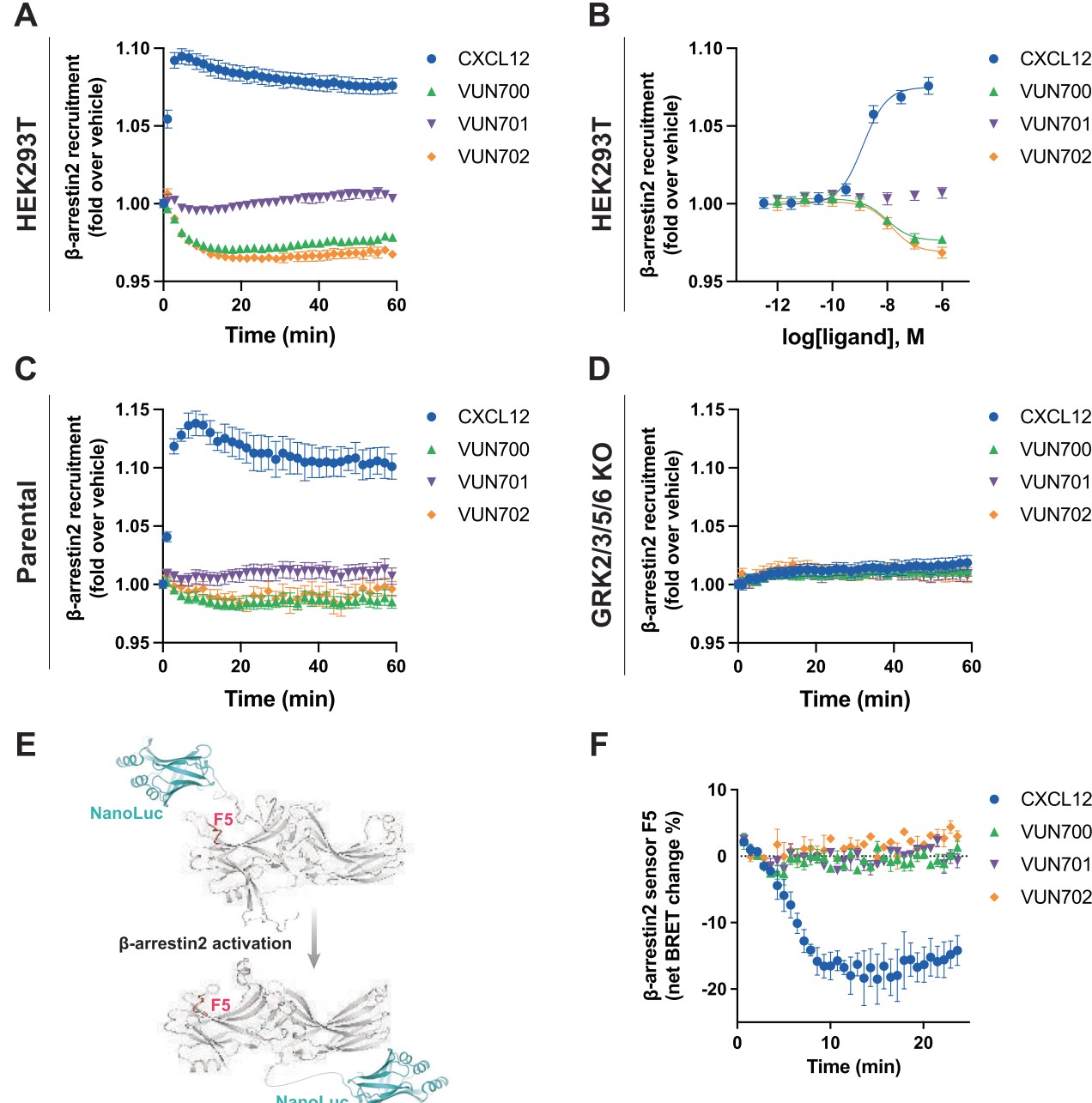

**Fig. 1 | ACKR3 nanobodies suppress basal β-arrestin2 interaction.**
**A, B** Recruitment of β-arr2-mV to ACKR3-Nluc measured by BRET (**A**) Time-dependent change in BRET over 60 min with either 316 nM of CXCL12 (blue circle) or 1 μM of nanobody (VUN700 (green triangle), VUN701 (purple inverted triangle), VUN702 (yellow diamond)) and (**B**) dose response curves of CXCL12 or nanobodies at 60 min recorded at 37 °C in HEK293T cells. **C, D** β-arrestin2 recruitment measured by BRET in agonist mode between donor ACKR3-Nluc and β-arr2-mV in (**C**) parental HEK293 or in (**D**) GRK2/3/5/6 KO HEK293 cells. **E** Schematic illustration of BRET-based FlAsH-tagged (CCPGCC) sensor F5 (between residues 156 and 157) on β-arrestin2. **F** Time-resolved changes in the NanoBRET β-arrestin2 conformational biosensor F5 signal upon ACKR3 activation, following the addition of 316 nM of CXCL12 or 1 μM of VUN700, VUN701, or VUN702 at 37 °C in parental HEK293 cells. Data are shown as the average ± SD of three independent experiments performed in technical triplicates.

rates provides insights into changes to the protein conformational state and protein-binding interfaces. This technique has recently been optimized for ACKR3 to monitor its interaction with small molecules[34].

Using differential HDX (ΔHDX) analyses, we compared the unbound (apo) and nanobody-bound states of ACKR3 (Fig. 2A, Supplementary Fig. 5 and 6, Supplementary Data 1 and 2). It is important to note that due to limited protein yield, the apo receptor reference was not derived from the same biological preparation for all nanobody conditions. To account for the inherent variability between membrane protein preparations, each nanobody condition was compared to its

corresponding apo. Despite these variations, the qualitative trends in ΔHDX remained consistent across replicates. Binding of all nanobodies protected the extracellular face from deuteration, confirming the nanobody binding interface proposed from CXCL12 competition assays (Supplementary Fig. 1). Differences in deuteration were localized to peptides corresponding to the orthosteric (CXCL12) binding pocket at the N-terminus (N-term, residues 27-33) and TM5 (residues 204-211) which displayed large protection upon binding (respective ΔHDX of up to 15% and 30% for each nanobody) (Supplementary Fig. 5B). The nanobodies significantly protected peptides in the N-terminus and

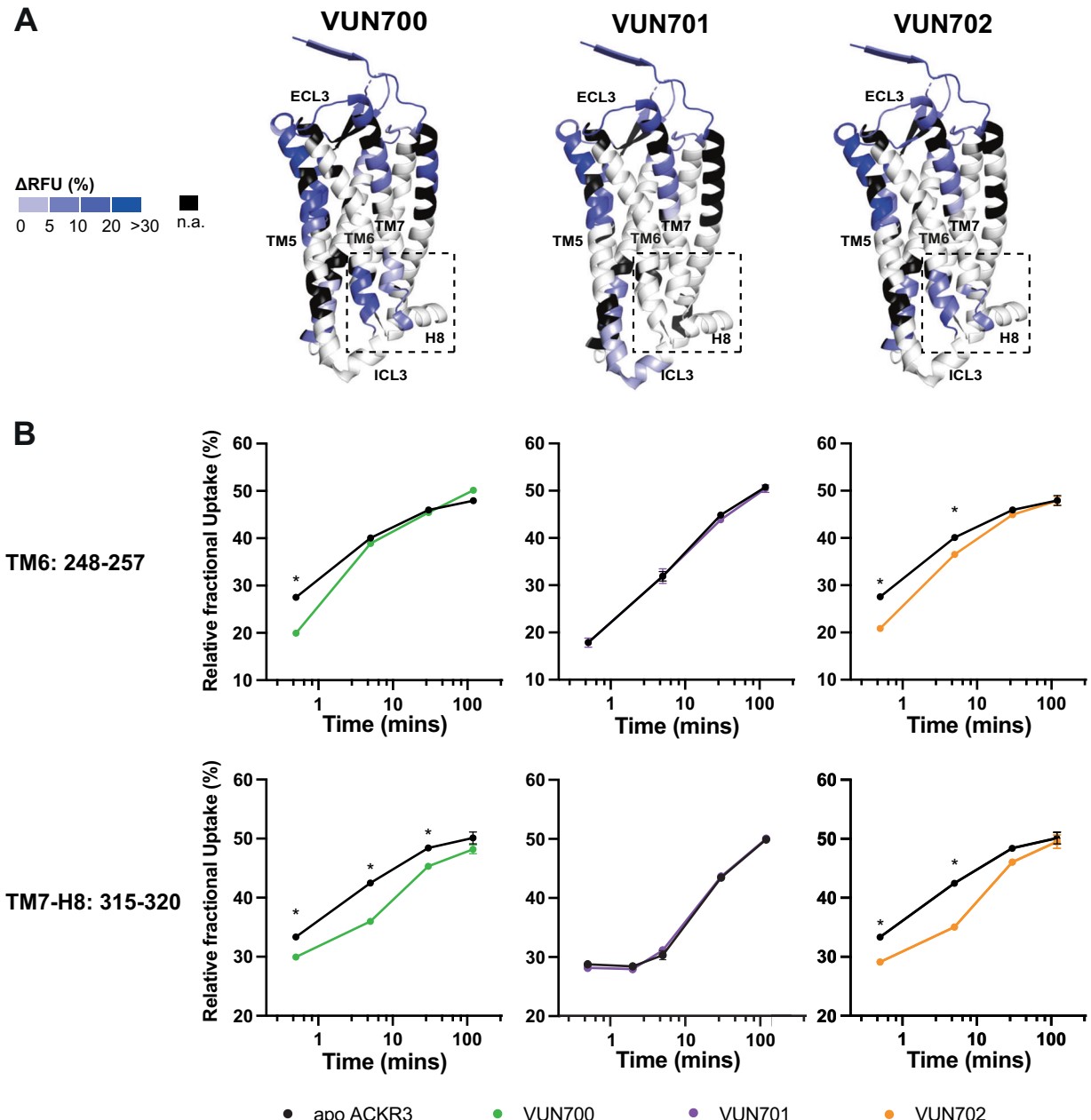

**Fig. 2 | Conformational changes in ACKR3 induced by nanobody binding.**
**A** Structural representation of the % differential relative fractional uptake (ΔRFU) data (apo ACKR3 – Nb-bound ACKR3) mapped onto the cryo-EM structure of ACKR3 (PDB:7SK5)[14]. This depicts reproducible and statistically significant ΔHDX over 120 minutes deuteration. The degree of ΔHDX (% ΔRFU) ΔRFU is represented according to the color scale. Black regions represent those with no sequence coverage. **B** Deuterium uptake plots showing time-dependent change in RFU for ACKR3 peptides on the intracellular side upon nanobodies binding, compared to apo ACKR3 (in black). Uptake represents the average and SD of three technical replicates from one biological preparation of ACKR3. Data is representative of three biological replicates. Statistically significant changes were determined using Deuteros 2.0 software[99] and Welch's t test (*, $p \leq 0.0005$).

extracellular loops (ECLs) of ACKR3 that correspond to the CXCL12 binding interface[14], suggesting that the nanobodies bind similarly to these receptor regions. More specifically, the N-terminal region spanning residues 25-33 exhibited reproducible and significant protection in the presence of the nanobodies, but only at short deuteration time points. This suggests that the nanobodies induce a local dynamic structuring in this region, which remains unstructured in the apo form.

While the overlapping interacting sites between CXCL12 and the nanobodies with ACKR3 explain their competitive binding mode, nanobody binding also induced conformational changes to the intracellular side of the receptor (Fig. 2B). It is known that the position of the cytoplasmic ends of TM6 and TM7 reflect the active state of GPCRs including ACKR3[14,34,35]. Only slight differences were observed for these

regions with the neutral antagonist VUN701 bound. In contrast, both inverse agonists VUN700 and VUN702 showed robust protection at the intracellular face of TM6, residues $248^{6 \times 31}$ to $257^{6 \times 40}$ (GPCRdb nomenclature in superscript[59]). Likewise, the inverse agonists induced protection at the linker region connecting TM7 to H8, residues $315^{7 \times 53}$ to $320^{8 \times 48}$, whilst VUN701 did not (Fig. 2B). These differences in HDX protection suggest that the inverse agonism observed for VUN700 and VUN702 is due to the promotion of an inactive ACKR3 conformation, while the neutral antagonist VUN701 does not impact the basal state of the receptor. Both results are consistent with the biological responses observed in Fig. 1.

To further structurally substantiate the ACKR3 conformational changes induced by the inverse agonist and antagonist nanobodies,

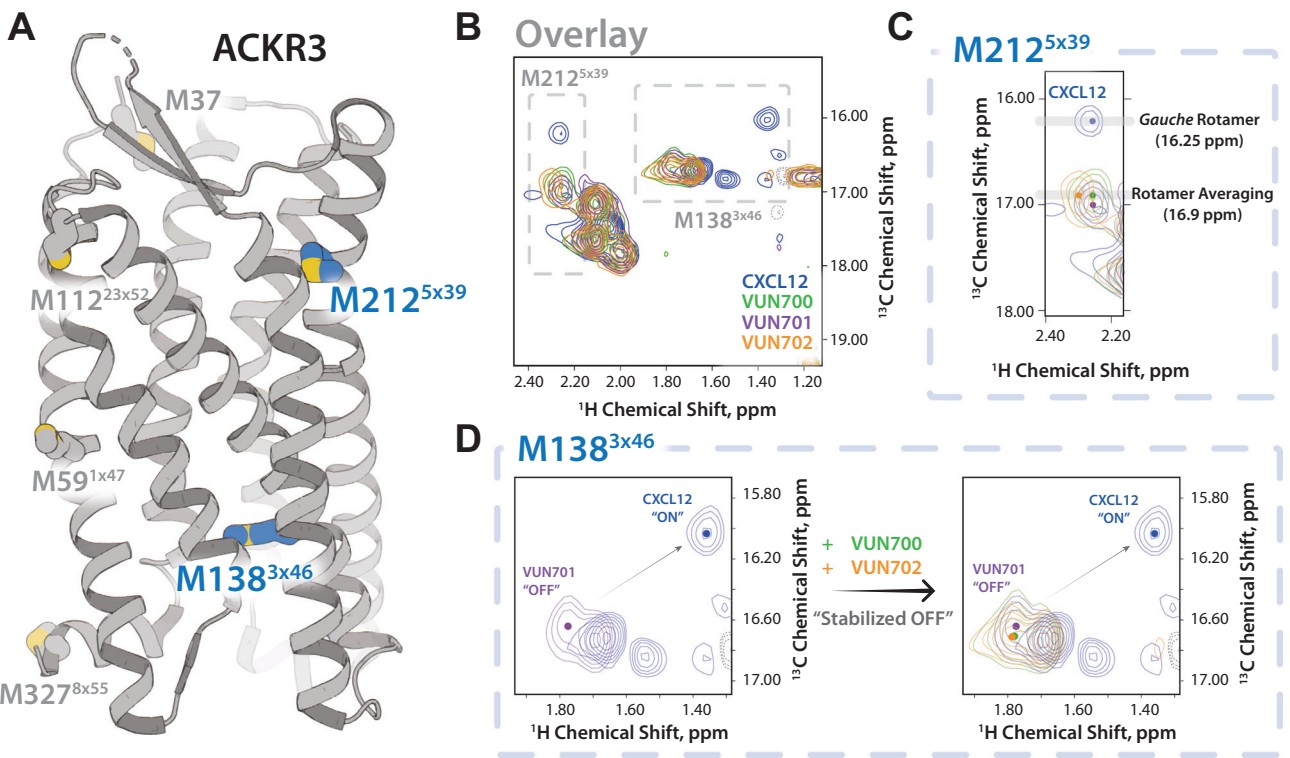

**Fig. 3 | NMR-based structural characterization of ACKR3 upon nanobodies VUN700 and VUN702 binding reveals a relatively more pronounced "OFF" state of ACKR3 than VUN701-bound state. A** ACKR3 structure (7SK6 PDB)[14] with NMR peaks M138[3x46] and M212[5x39] depicted. **B** $^1$H-$^{13}$C HSQC spectra of $^{13}$C-e-methionine labeled ACKR3 bound to the unlabeled agonist CXCL12 (blue), antagonist nanobody VUN701 (purple), or either inverse agonist nanobody VUN700 or VUN702 (green and yellow, respectively). **C** Overlay of M212[5x39] peaks from all ACKR3 complexes. **D** Overlay of M138[3x46] peaks from all ACKR3 complexes. The upfield peak positions (1H: -1.3 ppm) of M138[3x46] among agonist-bound states supports ring-current shifts due to aromatic side chain interactions.

$^{13}$CH$_3$-e-Met labeled ACKR3 was purified and analyzed by NMR spectroscopy[60]. As previously described[49], two of ACKR3's eight native methionines (M212[5x39] and M138[3x46], Supplementary Fig. 7) can be used to track receptor conformational dynamics at the ligand binding site and in the intracellular region, respectively (Fig. 3A). At first glance, the NMR analysis of ACKR3 bound to VUN700 or VUN702 results in similar spectra to that of ACKR3 bound to the neutral antagonist VUN701 (Fig. 3B). However, overlaying the spectra from the different ACKR3-nanobody complexes does reveal subtle shifts in the peaks for M212[5x39] (Fig. 3C) and M138[3x46] (Fig. 3D). Upon CXCL12 binding, the M212[5x39] position was previously shown to be in a gauche rotameric state with a peak at 16.25 ppm. This shifted downfield to 17.0 ppm with the neutral antagonist VUN701[49]. Binding of VUN700 or VUN702 further altered the M212[5x39] peak as compared to the ACKR3-VUN701 complex (Fig. 3C). In all three complexes, the $^{13}$C position of ~17.0 ppm is consistent with rotamer averaging and the absence of stabilizing interactions at the M212[5x39] position. In contrast, the M138[3x46] position showed a slight downfield shift in the $^{13}$C and $^1$H dimensions for VUN701, compared to the inverse agonists (Fig. 3D). Given the previous evidence that M138[3x46] exists as a mixture of active and inactive states, the shift along this line suggests that VUN700 or VUN702 binding shifts the ACKR3 conformational equilibrium relative to VUN701, potentially indicating a more "OFF" or inactive state of the receptor (Fig. 3D). Together with the HDX analysis, this NMR-based analysis suggests a distinct conformational state of the inverse agonist-bound ACKR3 compared to the antagonist-bound state.

### Inverse agonistic nanobodies trap ACKR3 at the plasma membrane

Constitutive internalization of ACKR3 contributes to chemokine scavenging[4,7,24,61] and is independent of receptor phosphorylation[5]. Therefore, we examined whether the inverse agonism displayed by

the VUN700 and VUN702 nanobodies elicited functional consequences on receptor trafficking from the plasma membrane (PM) to the early endosomes. First, we examined how the different nanobodies modulate ACKR3 internalization by monitoring the presence of ACKR3 at the PM with flow cytometry (Fig. 4A). CXCL12 internalized 25% of ACKR3 after 15 min exposure of cells and after 45 min, ACKR3 returned back to basal levels. In contrast, all nanobodies induced an increased level of ACKR3 on the membrane over time. The inverse agonists VUN700 and VUN702 increased the receptor level on the membrane by ~70% after 60 min of incubation. Interestingly, the neutral antagonist VUN701 also increased membrane presence of ACKR3, but to a lesser extent (~50%) and with a delay (Fig. 4A, Supplementary Fig. 8). We then investigated the subcellular trafficking of ACKR3 upon nanobody binding by employing BRET between ACKR3 and two different localization markers, mVenus-CAAX (mV-CAAX) for the plasma membrane and Rab5a-mVenus (Rab5a-mV) for the early endosomes[62] (Fig. 4B, C). Upon CXCL12 binding, ACKR3 rapidly internalized away from the plasma membrane (Fig. 4B) and appeared in the early endosomes (Fig. 4C). All three nanobodies inhibited constitutive internalization, causing the receptor to be retained at the membrane, consistent with the flow cytometry results. The accumulation is on a short enough time scale to suggest that the newly trapped receptors were previously constitutively internalized rather than de novo protein synthesis as ACKR3 shows little change in surface levels with hours of cycloheximide treatment[24,63]. This was more prominent for the inverse agonists VUN700 and VUN702 than for neutral antagonist VUN701 (Fig. 4B). Similarly, VUN700 and VUN702 impaired the basal trafficking of ACKR3 to the early endosomes, while VUN701 had little effect (Fig. 4C). The robust trapping of ACKR3 at the plasma membrane without depletion from early endosomes by VUN701 suggests that ACKR3 may be trafficked differently depending on receptor

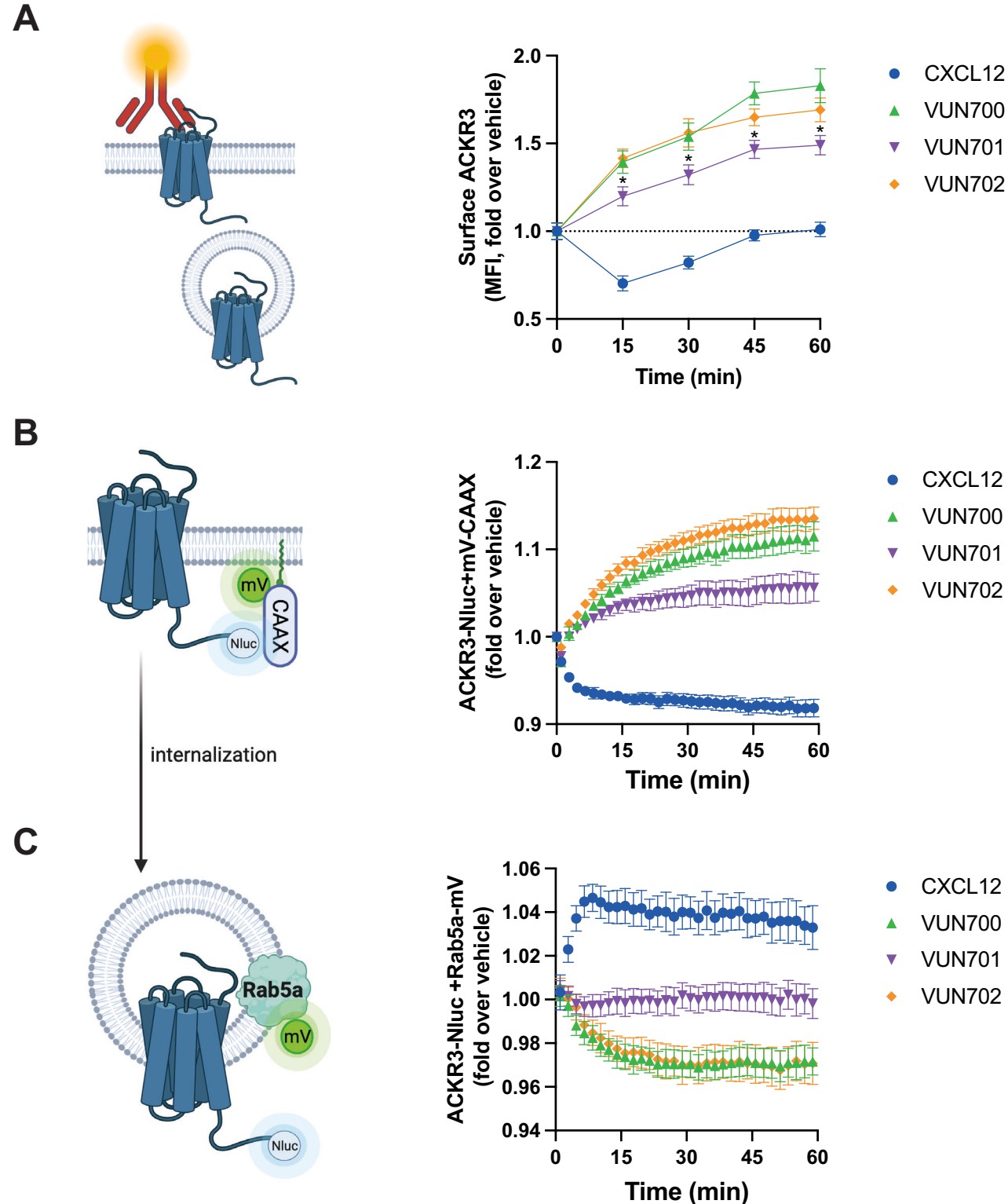

**Fig. 4 | ACKR3 nanobodies differentially change the localization of ACKR3 by capturing ACKR3 at the membrane. A** Surface ACKR3 detected by flow cytometry upon 316 nM of VUN700 (green triangle), VUN701 (purple inverted triangle) or VUN702 (yellow diamond), or 100 nM CXCL12 (blue circle) over 60 min at 37 °C in HEK293 cells. **B** Time-dependent internalization measured by BRET, between donor ACKR3-Nluc with mV-CAAX over 60 min with either 316 nM of CXCL12 or 1 μM of VUN700, VUN701 or VUN702 at 37 °C in HEK293T cells. **C** Time-dependent

ACKR3 localized in the early endosomes measured by BRET, between donor ACKR3-Nluc with Rab5a-mV, over 60 min with either 316 nM of CXCL12 or 1 μM of VUN700, VUN701, or VUN702 at 37 °C in HEK293T cells. Data is shown as the average ± SD of three independent experiments for (**A**) and four independent experiments for (**B**, **C**). performed in duplicate or triplicate. One-way ANOVA, multiple comparisons Dunnett test (* < 0.05). Cartoons generated in biorender.

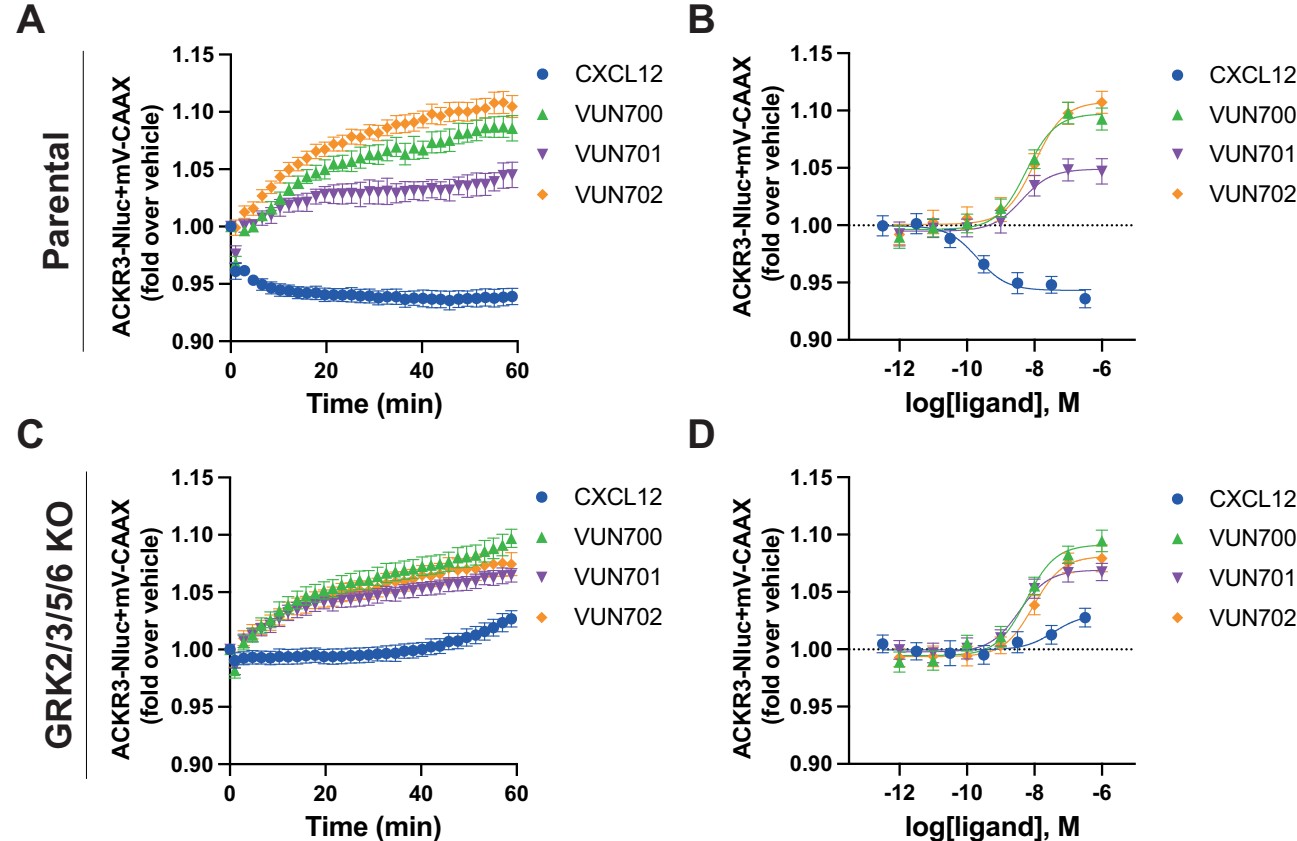

**Fig. 5 | ACKR3 nanobodies mediate GRK-independent and dependent constitutive internalization. A–D** Internalization of ACKR3-Nluc measured by BRET to the PM probe mV-CAAX in (A-B) parental HEK293 or in (C-D) GRK2/3/5/6 KO HEK293 cells at 37 °C. **A**, **C** Time-dependent change in BRET over 60 min with either 316 nM of CXCL12 (blue circle) or 1 μM of VUN700 (green triangle), VUN701 (purple inverted triangle), or VUN702 (yellow diamond) and (**B**, **D**) dose response curves of CXCL12 or nanobodies at 60 min. Data is shown as the average ± SD of four independent experiments performed in triplicate.

conformation and the neutral antagonist prevents a separate internalization mechanism apart from the early endosomal pathway. Taken together, nanobody binding blocks receptor intracellular trafficking by retaining the receptor on the membrane.

### All nanobodies inhibit GRK-independent internalization of ACKR3

CXCL12-mediated internalization of ACKR3 is β-arrestin-independent[6] but GRK-dependent[5], while constitutive internalization is independent of both effectors[5]. To determine if the nanobodies also impacted phosphorylation-independent internalization, the plasma membrane presence of ACKR3 was observed by BRET in GRK2/3/5/6 KO HEK293 cells (Fig. 5). In the parental cells, containing all GRKs, CXCL12 and the nanobodies showed the same order of effectiveness as shown in the HEK293T cells (Figs. 4B, 5A), with the inverse agonists VUN700 and VUN702 inducing greater redirection to the plasma membrane than the antagonist VUN701 (Fig. 5A, B). In the absence of GRKs, the internalization response from CXCL12 treatment is abolished, consistent with previous reports[5,55]. Unexpectedly, the inverse agonist and antagonist nanobodies elicited nearly identical levels of receptor retention in the membrane in the absence of GRKs, in contrast to the differential effects observed in the presence of GRKs (Fig. 5C vs 5D). The plasma membrane trapping effect by the nanobodies was independent of arrestins (Supplementary Fig. 9). These results suggest that constitutive internalization by ACKR3 can be divided into a phosphorylation-dependent component, which is suppressed by inverse agonism, and a phosphorylation-independent mechanism that is inhibited by both types of nanobodies tested here.

### Basal motility of cancer cells is reduced by ACKR3-directed nanobodies

ACKR3 is reported to contribute to cancer cell migration[64–66], but not due to activation by CXCL12[23,67]. Instead, we hypothesize that the constitutive activity of ACKR3 might play a role in cell motility. To resolve the influence of ACKR3 on non-chemokine driven migration, the basal or random movement of cervical cancer HeLa cells was tracked by live-single cell microscopy. HeLa cells express both ACKR3 and CXCR4 endogenously. This allows for examination of potential roles for ACKR3 in a relevant cellular context and in the presence of CXCR4. The cells displayed considerable motility in the presence of 10% FBS even without chemotactic stimulation (Fig. 6A). This basal motility was reduced when treated with VUN700 (Fig. 6A, B) showing a 30% average decrease in accumulated distance traveled with inverse agonist treatment. Inclusion of 10% FBS did not affect the inverse agonistic properties of VUN700 in the β-arrestin recruitment assay (Supplementary Fig. 10). To confirm that the effects are due to ACKR3 specifically, the receptor was genomically-knocked out from these cells by CRISPR (ACKR3-KO) (Supplementary Fig. 11). Without ACKR3, the cells showed reduced motility compared to WT HeLa cells, which showed no further reduction with the treatment with a nanobody, thereby suggesting the effect is due specifically to ACKR3 targeting. Similar results were observed when metastatic breast cancer MDA-MB-231 cells, which express high levels of endogenous ACKR3 and CXCR4[68], were treated with the ACKR3-targeting nanobodies (Supplementary Fig. 12). Basal motility of these cells was also reduced with VUN700 treatment, with a similar ~30% decrease in accumulated distance In both cancer cell lines VUN701 also affected basal motility (Fig. 6, Supplementary Fig. 12), which may be explained by long-term

treatment associated with trapping ACKR3 on the plasma membrane, seen for all ACKR3 nanobodies over time (Fig. 4A). VUN400, a CXCR4 targeting nanobody that inhibits CXCL12 binding and CXCL12-induced chemotaxis[45], had no effect on basal MDA cell motility. These results suggest a role for the basal activity of ACKR3 in mediating non-chemotactic movement of cancer cells.

## Discussion

ACKR3 is an atypical receptor that is best described as a chemokine scavenger. Although the receptor is implicated in many other physiological responses, they have not been explicitly tied to chemokine-mediated receptor activation or ligand scavenging. Here we present facets of ACKR3 constitutive activity with downstream responses using antagonistic and inverse agonistic ACKR3 nanobodies. The nanobodies had profound effects on the basal receptor events. While only the inverse agonistic nanobodies lead to a disruption of the arrestin-apo-ACKR3 complex, both inverse agonists and the antagonist suppressed constitutive internalization and trapped the receptor on the plasma membrane (Figs. 4, 7). These effects appear to be due to subtle changes in the conformational state of the receptor and may manifest into attenuation of basal, or random, cellular migration. These data provide insight into hidden functions of the atypical receptor that are independent of chemokine receptor activation.

Outward movement of TM6 is a common hallmark of GPCR activation[69,70] and ACKR3 is no exception[14,34,35]. The atypical receptor also displays extensive constitutive activity[14] and readily adopts an active conformation in the absence of stimulation[35]. This constitutive activity drives basal GRK phosphorylation and subsequent β-arrestin engagement (Fig. 1). VUN700 and VUN702 act as inverse agonists to suppress basal β-arrestin engagement, while the previously characterized VUN701[49] only blocked CXCL12-induced interactions (Fig. 1). This suggests that different conformations are being stabilized by the nanobodies, leading to different effects on ACKR3 phosphorylation and arrestin interactions. Indeed, both HDX and NMR studies reveal distinct conformations promoted by the inverse agonists compared to the antagonistic nanobody (Figs. 2, 3). The protection observed at the cytoplasmic ends of TM6 and TM7 (Fig. 2B) as well as the chemical shifts of M138$^{3x46}$ and M212$^{5x39}$ (Fig. 3B, C) are consistent with inactive receptor states. The inverse agonists VUN700 and VUN702 appear to stabilize an even more inactive conformation than VUN701 (Figs. 2, 3). These structural observations are in line with the more profound effects observed for the inverse agonistic nanobodies on ACKR3-arrestin engagement and membrane localization compared to the antagonist. In addition to scavenging ligands to regulate canonical receptor function, ACKR3 outcompetes CXCR4 for arrestins[71]. This is thought to protect CXCR4 from downregulation and desensitization due to overstimulation by CXCL12[4,71,72]. By releasing arrestins basally engaged with ACKR3, while also blocking activation by CXCL12, the inverse agonists could provide a tool to indirectly target CXCR4 or CXCR3 for downregulation.

All three nanobodies had profound inhibitory effects on the constitutive internalization of ACRK3 (Fig. 4). The atypical receptor undergoes both agonist-promoted internalization which is dependent on GRK phosphorylation as well as a 'passive' GRK-independent cycling between the plasma membrane and endosomes[5]. This second mechanism of receptor turnover is observed in other chemokine receptors[73] and contributes to chemokine scavenging, but is insufficient to fully replace the active internalization response[4]. Thus, even if co-binding with CXCL12 were possible, the nanobodies would prevent chemokine scavenging by inhibiting internalization of the receptor. The nanobodies retain the receptor at the plasma membrane, with the inverse agonists exhibiting significantly greater effects than the neutral antagonist in WT HEK293 cells (Fig. 5A, B). The nanobodies also re-directed ACKR3 to the plasma membrane in GRK2/3/5/6 KO cells (Fig. 5C, D), suggesting that even the neutral antagonist impacts the previously described passive internalization. These results

show that ACKR3 constitutive internalization occurs through both a GRK-dependent pathway, which requires receptor constitutive activation, and a GRK-independent pathway, operating via a heretofore undescribed mechanism. ACKR3 internalization does not require arrestins, which suggests a clathrin-independent mechanism, potentially through coordination via adenosine diphosphate ribosylation factors (ARFs), which regulate internalization and recycling pathways[74]. Alternatively, local membrane domains with particular lipid composition or curvature could sort GPCRs and mediate endocytosis during the natural turnover of the plasma membrane[75–77]. Thus, stabilization of ACKR3 by nanobody binding may segregate the receptor away from membrane regions primed to internalize, thereby leading to the trapping effect we observe.

The effects of VUN700 and VUN701 on the basal motility of HeLa and MDA-MB-231 cancer cells suggest that the constitutive activity of ACKR3 is implicated in migratory signaling (Fig. 6, Supplementary Fig. 12). The degree of inhibition was greater for VUN700 than VUN701 in agreement with the extent to which the two nanobodies inactivate receptor signaling. Moreover, the basal motility of HeLa cells was severely impaired upon knockout of ACKR3 (Fig. 6). The cells tested also express CXCR4, and an attractive explanation for the inhibition might be that the cells secrete CXCL12 and the balance of chemokine scavenging by ACKR3 is needed for CXCR4-mediated migration. However, blocking CXCR4 with VUN400[45] had no impact on the basal migration of these cells. Taken together, these findings imply that that inhibition of CXCL12 scavenging is not the mechanism for the impaired migration with ACKR3 nanobodies. We propose a chemokine-independent role for ACKR3 in basal motility of cancer cells. However, further studies are required to unravel how exactly and by which signaling pathways ACKR3 affects cell motility.

Biologics, including nanobodies, constitute an increasing proportion of FDA-approved therapeutics[47,78,79] The inverse agonist nanobodies developed in this study may possess several features with therapeutic potential. Besides their antagonistic properties, the ability of these nanobodies to shift the receptor into an inactive conformation may also prevent crosstalk between ACKR3 and interacting proteins and receptors (like CXCR4, EGFR, Cx43[3,80,81]). As noted above, ACKR3 protects CXCR4 from desensitization[4]. Such a mechanism could have important implications for targeting this axis in cancer therapeutics. A dual targeted approach could on one hand antagonize CXCR4 while also downregulate the canonical receptor via ACKR3 inverse agonism. ACKR3 forms a complex with the gap junction protein Cx43 upon CXCL12 activation in astrocytes[3]. The atypical receptor coordinates the internalization of Cx43 in a β-arrestin-dependent manner, which inhibits gap intercellular communication. The inverse agonistic nanobodies could therefore preserve Cx43 on the plasma membrane and protect these structures. Knowing the capabilities of the current inverse agonists, expanding their modulatory activity through structural engineering would be of interest. Specific engineering of VUN701 has already successfully converted the antagonistic nanobody into an agonist[54]. Nanobody engineering also generated a universal platform to support structural determination of membrane proteins[82]. New computational methods for designing nanobodies targeting specific epitopes with high affinity binders will continue to expand possible applications[83,84]. Alternatively, recently reported small molecule inverse agonists provide another avenue for targeting the constitutively active ACKR3 state[39,40]. Further studies are necessary to ascertain whether differentially modulating ACKR3 is essential and/or beneficial when targeting ACKR3-related diseases including cardiovascular diseases (as atherosclerosis)[85], autoimmune diseases (as multiple sclerosis)[86,87], and cancer[64,88].

In summary, we have identified a basal state of ACKR3 that displays constitutive activity and is involved in cancer cell motility. We investigated this through nanobodies binding to the extracellular site of ACKR3 with distinct properties. The inverse agonistic properties of two of these molecules emphasize the constitutive activity of the

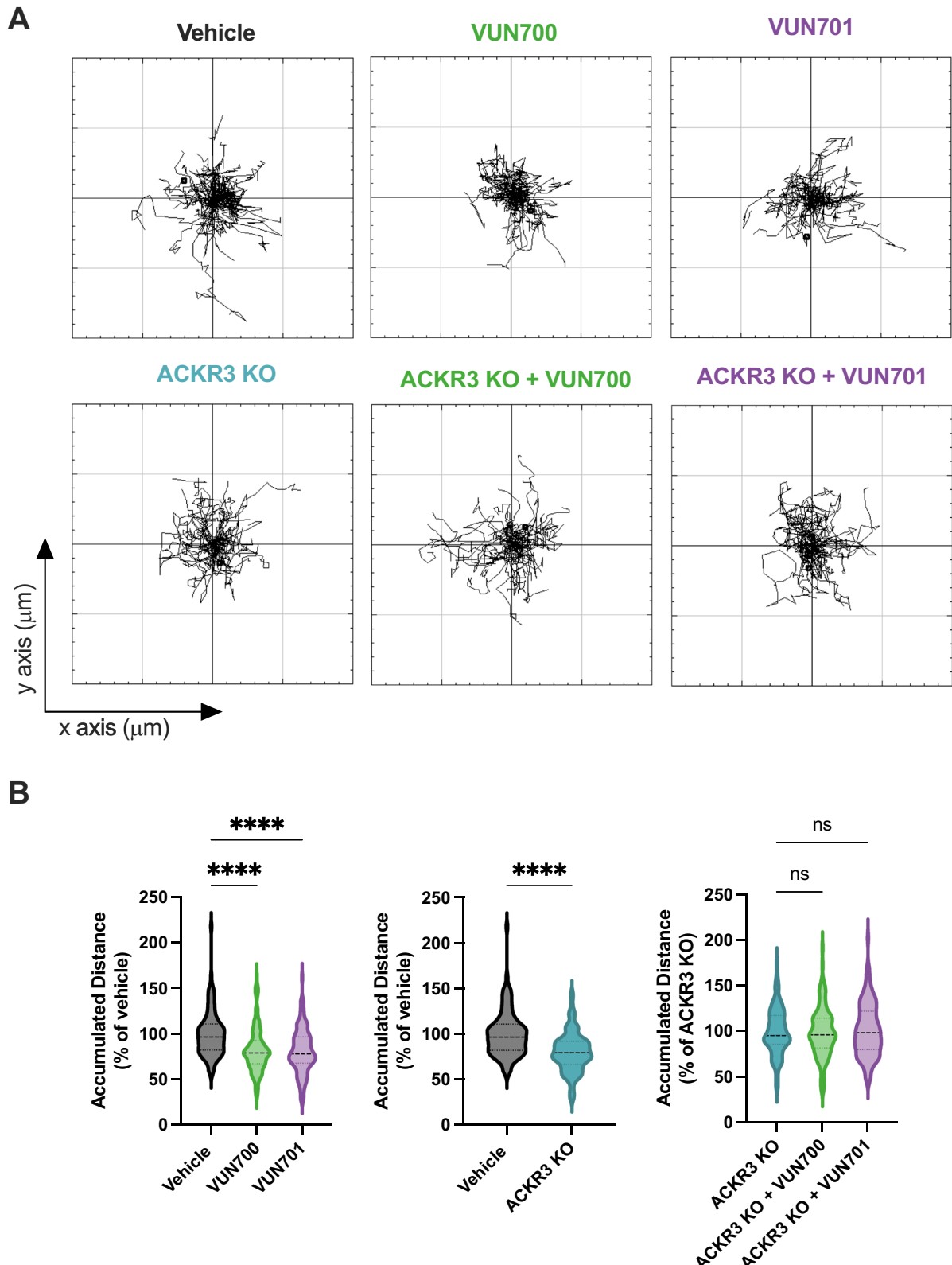

**Fig. 6 | Basal motility in HeLa cells is reduced upon treatment with nanobodies targeting ACKR3 as well as ACKR3 CRISPR knock-out. A** Representative basal motility paths of HeLa cervical cancer cells traced for 16 h in 10% FBS, vehicle condition (in black), treated with 100 nM of VUN700 (in green) or VUN701 (in purple). And representative basal motility paths of ACKR3 knock-out HeLa cervical cancer cells traced for 16 h in 10% FBS, vehicle condition (in gray), treated with 100 nM of VUN700 (in green) or VUN701 (in purple). **B** Accumulated distance (total distance traveled) of all individual cells from each of the conditions in (**A**). The positions of at least 120 cells, selected across three independent experiments ( ~ 40 cells/repeat), were tracked and normalized to the vehicle condition. Significance was determined by one-way ANOVA Dunnett test ($p < 0.0001$ (****)).

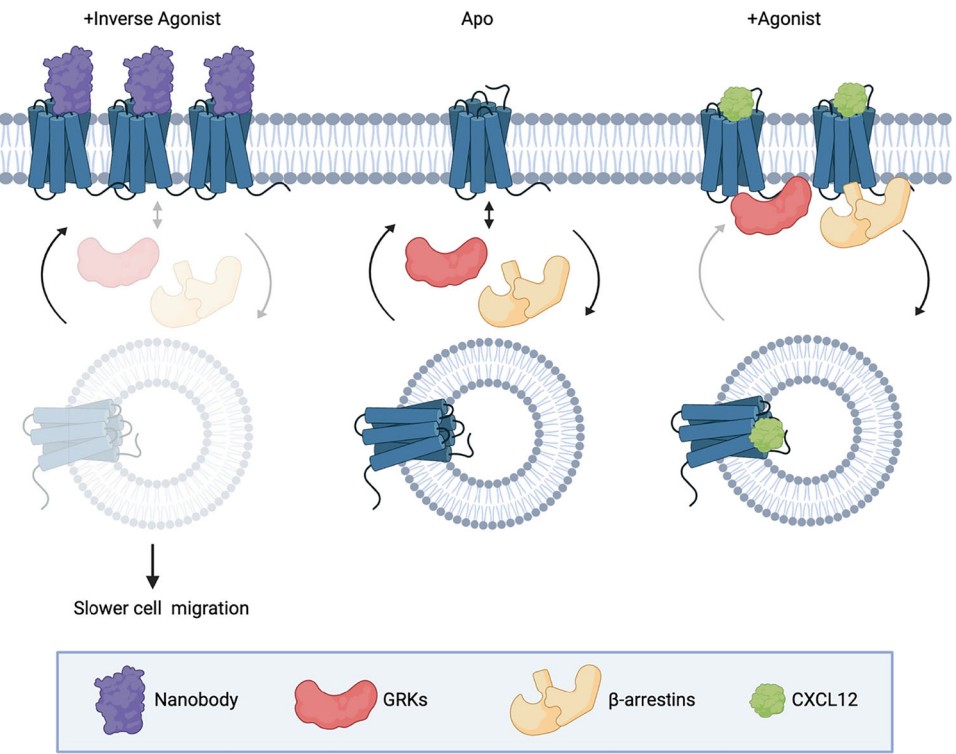

**Fig. 7 | Constitutive activity of atypical chemokine receptor revealed by inverse agonistic nanobodies.** Apo-ACKR3 is constitutively active, which leads to the receptor basal GRK phosphorylation, arrestin recruitment, and internalization. Stimulation with an agonist like CXCL12 fully activates ACKR3 and drives robust phosphorylation, arrestin complexing, and endocytosis. Inverse agonistic nanobodies suppress the constitutive activity of ACKR3, trapping the receptors at the cell surface and reducing interactions with arrestins or GRKs, thereby allowing GRKs and arrestins to be available for other GPCRs. These nanbodies also slow basal cell migration, suggesting a role for ACKR3 constitutive activity in cell motility. Figure generated in biorender.

receptor by impairing basal β-arrestin engagement as well as its constitutive internalization by stabilizing an inactive receptor conformation. Inhibition of basal ACKR3 activity attenuated basal cell motility, which reveals a distinct role for ACKR3 in cellular biology and cancer in particular. These results open new avenues and strategies for therapeutically targeting ACKR3.

## Methods

Unless stated otherwise, all chemicals and reagents were purchased from Melford or Sigma-Aldrich. ACKR3 nanobodies were provided by QVQ VUN700 (product number Q125), VUN701 (product number Q123) and VUN702 (product number Q126).

### Cell culture and transfection

Human embryonic kidney 293 T (HEK293T) and MDA-MB-231 breast cancer cells were obtained from ATCC. CHO cells were obtained from ThermoFisher. NanoLuc-CXCR4 and NanoLuc-ACKR3 CRISPR Knock In HeLa cells were a kind gift from Stephen Hill from University of Nottingham[89]. HEK293 Parental and CRISPR HEK293 β-arrestin1/β-arrestin2 Knock Out (KO) cells were provided by Asuka Inoue from Tohoku University. Stable HEK293 expressing SNAP-tagged ACKR3 were kindly provided by Meritxel Canals from University of Nottingham. HEK293 Parental and CRISPR GRK2/3/5/6 KO HEK293 cells were provided by Julia Drube and Carsten Hoffmann from University Hospital Jena. NanoLuc-ACKR3 HeLa, NanoLuc-CXCR4 HeLa, HeLa, HEK293T, HEK293, HEK293 β-arrestin1/2 KO, stable ACKR3 HEK293, and CHO cells were cultured in Dulbecco's Modified Eagle's Medium (DMEM, Thermo Fisher Scientific, Gibco, #41966) supplemented with 100 Units of penicillin, 100 g/mL streptomycin (Pen/Strep, Gibco, #15140-122) and 10% (v/v) Fetal Bovine Serum (FBS, Bodinco). ACKR3 stable cell line was maintained under antibiotic selection with G418 500 µg/ml (#A1720 Sigma-Aldrich).

HEK293T, HEK293, and HEK293 β-arrestin1/2 KO cells were transfected in suspension with a total of 1 µg DNA and 6 µg 25 kDa linear polyethyleimine (PEI, Polysciences Inc.) in 150 mM NaCl solution per 1 million cells. CHO cells were transfected in Opti-MEM reduced serum media (Thermo Fisher Scientific, Gibco, #51985-034) with identical DNA-PEI ratio. DNA encoding ACKR3 and biosensors was supplemented with empty pcDEF3 or pcDNA3 to obtain a total DNA amount of 1 µg. The DNA-PEI mixture was vortexed for 5 seconds and incubated for 15 min at room temperature (RT). Cells were detached with Trypsin (Gibco) and resuspended in DMEM. A $3 \times 10^5$ cells/mL cell suspension was added to DNA-PEI mixture and cells were seeded at a density of 30,000 in white flat-bottom 96-well plates (Greiner Bio-One) and incubated for 48 h.

### CRISPR ACKR3 knock out generation in HeLa cells

HeLa cells were transfected with 1 µg pX459 (sgRNA#4 sequence: gtgggttgtcctcaccatcc) and 4 µL Fugene 4 K (Promega). 24 h post transfection the cells were selected with 1.75 µg/mL puromycin. Selection was maintained until all cells in an un-transfected control well had died. The resistant cells were single-cell seeded in a 96-well plate to obtain one pure colony of HeLa ACKR3 KO cells. From the selected single-cell colonies the genomic DNA was isolated and the ACKR3 gene was amplified by PCR. The obtained ACKR3 gene fragments were then analysed by Sanger sequencing and compared to the parental ACKR3 gene fragment to confirm an indel mutation in the ACKR3 KO gene in both alleles.

### ACKR3 Nanobodies selection via phage-display

ACKR3 nanobodies VUN700, VUN701, and VUN702 and CXCR4 nanobody VUN400 were provided by QVQ. Their generation, including llama immunization, library construction, and nanobody selection were performed as described previously[45,90–92]. Briefly, two llamas were

immunized with ACKR3-encoding plasmid DNA, cDNA was generated from peripheral blood mononuclear cells, nanobody sequences were amplified by PCR, and nanobody phage display libraries were constructed. Selections for ACKR3-specific binders were performed using three consecutive rounds of phage panning on ACKR3-expressing or empty (null) virus-like particles (Integral Molecular, Philadelphia, PA, USA) immobilized in MaxiSorp plates (Nunc, Roskilde, Denmark).

## Nanobodies production

Nanobody-FLAG-6xHis proteins purification was performed as previously described[93]. pMEK222-transformed BL21 codon+ DH5alpha cells, respectively, were grown in LB/2% glucose O/N at 37 °C. The O/N preculture was inoculated in regular terrific broth and grown for 3 h at 37 °C, after which periplasmic expression of nanobodies was induced for another 4 h by addition of 1 mM isopropyl-b-D-thiogalactopyranoside (IPTG, Sigma-Aldrich) to the culture medium. After production, cultures were spun down for 30 min at 3500 × g and the pellets were frozen overnight at −20 °C. The next day, pellets were thawed and resuspended in PBS. The resuspended pellet was incubated at 4 °C head-over-head at 20 RPM for 2 h. Cultures were spun down for 30 min at 3500 × g at 4 °C and, after filtering using a 0.45 µM filter (VWR), the periplasmic fraction (supernatant) was stored at 4 °C until purification.

Nanobody-FLAG-6xHis proteins were purified using ROTI®Garose-His/Co Beads (Carl Roth GmbH & Co, DE). Samples were first eluted in a buffer containing 500 mM imidazole PBS (Sigma-Aldrich, St. Louis, MO, USA) and after dialyzed using Snakeskin Dialysis Tubing 10 kDa molecular weight cutoff (MWCO) membranes (Thermo Fisher Scientific) in PBS. Purity of all produced and purified nanobodies was verified by sodium dodecyl sulfate-polyacrylamide gel electrophoresis (SDS-PAGE) under reducing conditions.

## NanoBRET CXCL12 competition assay

30k cells per well of NanoLuc-ACKR3 or NanoLuc-CXCR4 Knock In (KI) HeLa cells were seeded in white flat-bottom 96-well plate. 24 h later, cells were washed with PBS once, and Hank's Buffered Saline Solution (HBSS) supplemented with 0.1% BSA, 10 mM HEPES, 1 mM MgCl$_2$ and 2 mM CaCl$_2$ was added to the cells. Subsequently, increasing concentrations of unlabeled CXCL12 (Almac) or unlabeled nanobodies were added as indicated in the figures, and incubated for 45 min at RT. Next, 3.3 nM CXCL12-AF647 (Almac) was added and incubated for 15 min at RT. Next, luciferase substrate (Furimazine, Nano-Glo® substrate (Promega, #N1110, final concentration of 15 µM)) was added and luminescence was measured using a PheraSTAR plate reader (BMG) with 460 ± 80 nm and 610-LP nm filters until the luminescence signal stabilized. BRET data were normalized to full homologous displacement (0%) and fluorescent CXCL12 only (100%). Data were analyzed with a nonlinear fit to create a dose-response curve in Graph Pad Prism Version 10.2.0. Data from all independent experiments were used in the analysis and calculation of standard deviation.

## NanoBRET assays

1 million cells were transfected with 2 µg total DNA, consisting of BRET donor, BRET acceptor, supplemented with empty plasmid. Position of genetically fused sensors is given when stating the construct (e.g., in Nluc-ACKR3 Nluc tag is located in the N-terminus of ACKR3, while ACKR3-Nluc, Nluc is on the C-terminus). In the β-arr1/2 recruitment and internalization experiments, 30–50 ng of HA-ACKR3 WT-Nluc was used in combination with 150–250 ng of the different acceptors, β-arr1/2-mVenus, mVenus-CAAX or Rab5a-mVenus (donor:acceptor ratio 1:5). Cells were transfected in suspension with PEI in a ratio of PEI µg:DNA ug 6:1, with a total amount of 2 µg of DNA per million cells. Cell suspension of 300k/ml is used to seed 30k/well cells in 96 well plate. 48 h after transfection, cells were washed with PBS once, and Hank's Buffered Saline Solution (HBSS) supplemented with 0.1% BSA, 10 mM HEPES, 1 mM MgCl2 and 2 mM CaCl2 was added to the cells. To assess whether FBS impacted nanobody activity in migration assays, HBSS was further supplemented with 10% FBS in the relevant conditions in the NanoBRET assay. Incubation with the luciferase substrate (Furimazine, Nano-Glo® substrate (Promega, #N1110, final concentration of 15 µM)) followed and the basal BRET was measured for 5 minutes. BRET for this pair Nluc and mVenus was measure at 460 ± 30 nm and 535 ± 30 nm respectively. Next, cells were treated with CXCL12 or nanobodies indicated in the legend of the figures and consecutively, measured BRET for 60 minutes. Normalized BRET ratio was then calculated by dividing the raw BRET values of each well from the ligand induced results by the basal BRET measured before stimuli (baseline). For vehicle normalization, normalized BRET value was divided by the vehicle condition (without ligand) over time, to normalize for effects of the drop in furimazine availability. Data were analyzed with a nonlinear fit to create a dose-response curve in GraphPad Prism. Data from all independent experiments were used in the analysis and calculation of the standard deviation.

## Intramolecular FlAsH-NanoBRET assays

β-arrestin2 conformational change biosensors used in this work were previously described[58]. CRISPR control cells were transfected with 1.2 µg of untagged ACKR3, 0.12 µg of a β-arrestin2 FlAsH-tagged biosensor C-terminally coupled to NanoLuc, and 0.25 µg of an empty vector, following the Effectene transfection reagent protocol by Qiagen. In total, three sensors were used, numbered as β-arrestin2 FlAsH-3,5 10-NanoLuc sensors. The following day, 40,000 cells were seeded per well into poly-D-lysine-coated 96-well plates and incubated overnight at 37 °C. For this study, the FlAsH (fluorescein arsenical hairpin-binder)-labeling procedure, previously described by Hoffmann et al.[94], was adjusted for 96-well plates. Briefly, the cells were washed twice with PBS, then incubated with 250 nM FlAsH or mock DMSO in labeling buffer (150 mM NaCl, 10 mM HEPES, 25 mM KCl, 4 mM CaCl$_2$, 2 mM MgCl$_2$, 10 mM glucose; pH 7.3), complemented with 12.5 µM 1,2-ethane dithiol (EDT) for 1 h at 37 °C. After aspiration of the FlAsH labeling and mock labeling solutions, the cells were incubated for 10 min at 37 °C with 100 µl of 250 µM EDT in labeling buffer, per well. The NanoLuc substrate was added and a basal measurement was recorded for 3 min. Subsequently, either CXCL12 or VUN700, VUN701, or VUN702 nanobodies were added in the required concentrations and BRET was measured for 20 minutes. Analysis of the BRET change was performed as described above (see Section "NanoBRET assays"). Measurements were performed using the Synergy Neo2-provided BRET2 filter (Emission wavelengths 400/510).

## Expression and purification of ACKR3 (HDX)

For production in insect cells, the full-length gene of human ACKR3 was subcloned into pFastBac1 to enable infection of Sf9 insect cells. The construct bore a hemagglutinin signal peptide followed by a Flag-tag preceding the receptor sequence. ACKR3 N13, N22 and N33 residues were substituted with a Glutamine in order to avoid N-glycosylation.

Flag-ACKR3 was expressed in sf9 insect cells using the pFastBac baculovirus system (Thermo Fisher Scientific). Cells were grown in suspension in EX-CELL 420 medium (Sigma-Aldrich) and infected at a density of 4 × 10$^6$ cells/ml with the recombinant baculovirus. Flasks were shaken for 48 h at 28 °C, subsequently harvested by centrifugation (3000 × g, 20 min) and stored at -80 °C until use. Cell pellets were thawed and lysed by osmotic shock in a buffer containing 10 mM Tris (pH 7.5), 1 mM EDTA, 2 mg/ml iodoacetamide, 1 µM ACKR3 agonist VUF11207 and protease inhibitors: 50 µg/ml Leupeptin (Euromedex), 0.1 mg/ml Bensamidine (Sigma-Aldrich) and 0.1 mg/ml Phenylmethylsulfonyl fluoride (PMSF; Euromedex). Lysed cells were centrifuged for (38,400 × g, 10 min) and the resulting pellet was solubilized and dounce-homogenized 20x in buffer containing 50 mM Tris (pH 7.5), 150 mM NaCl, 2 mg/ml iodoacetamide, 1 µM VUF11207, 0.5% n-dodecyl-β-maltoside (DDM) (Anatrace), 0.1% Cholesteryl hemisuccinate (CHS) and protease inhibitors (50 µg/ml Leupeptin, 0.1 mg/ml Bensamidine and 0.1 mg/ml PMSF). The homogenate

was subsequently stirred for 1 h at 4 °C and centrifuged (38,400 × $g$, 30 min). The supernatant was then loaded onto M2 anti-Flag affinity resin (Sigma-Aldrich) using gravity flow. Resin was subsequently washed with 10 column volumes (CV) of DDM wash buffer containing 50 mM Tris, 150 mM NaCl, 0.1 μM VUF11207, 0.1% DDM, 0.02% CHS. Detergent was then gradually exchanged from DDM to lauryl maltose neopentyl glycol (LMNG) (Anatrace) using increasing ratios of DDM wash buffer and buffer containing 50 mM Tris, 150 mM NaCl, 0.02 μM VUF11207, 0.2% LMNG, 0.05% CHS. Once detergent was fully exchanged, LMNG and CHS concentration were steadily reduced to 0.005% and 0.001% respectively. ACKR3 was finally eluted in 50 mM Tris, 150 mM NaCl, 0.02 μM VUF11207, 0.005% LMNG, 0.001% CHS and 0.4 mg/ml Flag peptide (Covalab). The eluate was concentrated using a 50 kDa MWCO concentrator (Millipore), then ACKR3 was purified by size exclusion chromatography (SEC) using a Superdex 200 Increase (10/300 GL column) connected to an ÄKTA purifier system (GE Healthcare) and eluted in buffer elution buffer without Flag peptide or VUF11207. Fractions containing monomeric ACKR3 were concentrated to between 20 and 25 μM, aliquoted, flash-frozen and stored at −80 °C prior to HDX experiments.

## HDX-MS experiments
HDX-MS experiments were performed using a Synapt G2-Si HDMS coupled to nanoACQUITY UPLC with HDX Automation technology (Waters Corporation). ACKR3 in LMNG detergent was concentrated up to 20–25 μM and optimization of the sequence coverage was performed on undeuterated controls. Various quench times and conditions were tested; in the presence or absence of different denaturing or reducing reagents, with or without longer trapping times to wash them out. The best sequence coverage and redundancy for ACKR3 were systematically obtained without the addition of any denaturing agents in the quench buffer. Mixtures of receptor and nanobody were pre-incubated to reach equilibrium prior to HDX-MS analysis. Analysis of freshly prepared ACKR3 apo, ACKR3:nanobody (1:2 ratio) were performed as follows: 3 μL of sample were diluted in 57 μL of undeuterated for the reference or deuterated last wash SEC buffer. The final percentage of deuterium in the deuterated buffer was 95%. Deuteration was performed at 20 °C for 0.5, 2, 5, 30 and 120 min. Next, 50 μL of reaction sample was quenched in 50 μL of quench buffer (50 mM KH₂PO₄, 50 mM K₂HPO₄, 200 mM TCEP pH 2.3) at 0 °C. 80 μL of quenched sample was loaded onto a 50 μL loop and injected on a Nepenthesin-2 column (Affipro) maintained at 15 °C, with 0.2% formic acid at a flowrate of 100 μL/min. The peptides were then trapped at 0 °C on a Vanguard column (ACQUITY UPLC BEH C18 VanGuard Pre-column, 130 Å, 1.7 μm, 2.1 mm × 5 mm, Waters) for 3 min, before being loaded at 40 μL/min onto an Acquity UPLC column (ACQUITY UPLC BEH C18 Column, 1.7 μm, 1 mm X 100 mm, Waters) kept at 0 °C. Peptides were subsequently eluted with a linear gradient (0.2% formic acid in acetonitrile solvent at 5% up to 35% during the first 6 min, then up to 40% and 95% over 1 min each) and ionized directly by electrospray on a Synapt G2-Si mass spectrometer (Waters). HDMSE data were obtained by 20–30 V trap collision energy ramp. Lock mass accuracy correction was made using a mixture of leucine enkephalin and GFP. For every tested condition we analyzed two to three biological replicates, and deuteration timepoints were performed in triplicates for each condition.

Peptide identification was performed from undeuterated data using ProteinLynx global Server (PLGS, version 3.0.3, Waters). Peptides were filtered by DynamX (version 3.0, Waters) using the following parameters: minimum intensity of 1000, minimum product per amino acid of 0.2, maximum error for threshold of 10 ppm. All peptides were manually checked, and data was curated using DynamX. Back exchange was not corrected since we are measuring differential HDX and not absolute. Statistical analysis of all ΔHDX data was performed using Deuteros 2.0 by applying a peptide-level significance test and only peptides with a 99% confidence interval were considered. All supporting data and meta-data are presented in Supplementary Data 1 (HDX summary tables) and 2 (HDX data tables).

## pFastBac constructs and mutant generation (NMR)
Human ACKR3 WT pFastBac1 plasmid was generously provided by Dr. Tracy Handel (UC San Diego). The construct is comprised of a gp64 promoter, N-terminal HA signal sequence for membrane localization, human WT ACKR3 (unmodified except with removal of the N-terminal methionine), a C-terminal PreScission protease cleavage tag, and FLAG / 10x His tags as described previously[95].

## Baculovirus preparation and ACKR3 expression (NMR)
Baculovirus generation and ACKR3 expression was performed as described previously[95,96]. Briefly, recombinant baculovirus was produced using the Bac-to-Bac Baculovirus Expression System (Invitrogen). A pFastBac1 plasmid containing the described ACKR3 construct was transformed into DH10Bac *E. coli* (Thermo Fisher) and subsequently plated onto LB agar with 50 μg ml⁻¹, kanamycin, 7 μg ml⁻¹ gentamicin, 10 μg ml⁻¹ tetracycline, 100 μg ml⁻¹ Bluogal and 40 μg ml⁻¹, IPTG (Teknova). Blue/white screening identified recombinant (white) colonies, of which individual clones were inoculated in 5 mL of LB with 50 μg ml⁻¹, kanamycin, 7 μg ml⁻¹ gentamicin, and 10 μg ml⁻¹ tetracycline and placed in a 37 °C shaking incubator overnight. Cultures were pelleted, lysed, and neutralized using buffers from the GeneJET Plasmid Midiprep Kit (Thermo Fisher) and bacmid was purified by isopropanol precipitation (see reference[95] for details). Final bacmid pellets were solubilized in 40 mM Tris, pH 8 and 1 mM EDTA. To transfect Sf9 cells a mixture of purified bacmid (5 μl–1 μg total DNA), X-TremeGENE HP DNA (3 μl) and Expression Systems Transfection Medium (100 μl) was mixed and added to 2.5 ml of Sf9 cells at ~ 1.2 × 10⁶ cells ml⁻¹ and the bacmid-cell mixture was placed in a 24-well, deep-well plate (Thomson Instrument Company) covered with a polyurethane sealing film (Diversified Biotech). Cells were incubated at 27 °C for 96 h at 300 rpm. Cells were subsequently pelleted, and the supernatant was collected to isolate P0 ("zero passage") virus. P0 virus titers were determined using gp64 titer assay as described previously[95]. Next, P1 ("first passage") virus was produced by infecting 50 ml of Sf9 cells at a density of ~ 2.0 × 10⁶ cells ml⁻¹ with titered P0 virus at an MOI of 0.1-0.5. Cells were incubated at 27 °C for 72 h shaking at 144 rpm. Cells were pelleted and the supernatant was collected and titered using the same gp64 titering assay[95]. Large scale expression of labeled ACKR3 was performed by adding high titer P1 virus (≥ 1 × 10⁻⁹ IU/ml) to ~2 L of Sf9 cells in methionine deficient medium (Expression Systems) at a density of 3.5 – 4.0 × 10⁶ cells ml⁻¹ at an MOI of 5[96]. After 5 h post-infection, 250 mg L⁻¹ ¹³CH₃-methionine (Cambridge Isotope Laboratories) was added to cells. Cells were incubated at 27 °C for 48 h then pelleted and stored at −80 °C.

## ACKR3 purification (NMR)
Receptor was purified described previously[96] with some modifications. Briefly, frozen cell pellets (~50 ml frozen cells per 2 L of cell culture) were diluted 1:1 in hypotonic buffer (10 mM HEPES pH 7.5, 20 mM KCl, 10 mM MgCl₂, Roche Complete Protease Inhibitor Cocktail, 2 mg ml⁻¹ iodoacetamide) and thawed on ice. Direct solubilization was performed by passing the cell slurry through a 16-gauge needle four times to aid solubilization by lysing cells. Cell slurry was added to 1:1 to solubilization buffer (100 mM HEPES pH 7.5, 800 mM NaCl, 1.5% / 0.3% LMNG/CHS) and incubated at 4 °C for 4 h with stirring. The mixture was spun down at 50,000 × g for 30 min and the supernatant (~200 ml) was transferred to 4 × 50 ml conical tubes. 4 ml TALON cobalt resin slurry (Takara Bio Inc.) was added (1 ml/tube) with 10 mM imidazole final concentration to limit non-specific binding, and the supernatant mixture was rocked overnight at 4 °C. This mixture was added to columns the following day, and cobalt resin was washed with 20 ml of two wash buffers (Wash Buffer 1: 50 mM HEPES pH 7.5, 400 mM NaCl, 0.1%/ 0.02% LMNG/CHS, 10% glycerol, 20 mM imidazole; Wash Buffer 2: 50 mM HEPES pH 7.5, 400 mM NaCl, 0.025%/0.005% LMNG/CHS, 10% glycerol, 10 mM imidazole). ACKR3 was eluted with a high imidazole buffer (50 mM HEPES pH 7.5, 400 mM NaCl, 0.025%/0.005% LMNG/

CHS, 10% glycerol, 250 mM imidazole). Elutions were concentrated to 500 μl using a 30,000 MWCO Amicon Ultra-4 Centrifugal Filter Unit (Millipore Sigma) and buffer exchanged into Exchange Buffer (25 mM HEPES pH 7.5, 150 mM NaCl, and 0.025%/0.005% LMNG/CHS) using a PD-10 desalting column (GE). Precission Protease and PNGaseF were added to purified ACKR3 overnight. The next day, 500 μl TALON cobalt resin was added, and the mixture was incubated with rocking for 2 h at 4 °C. The mixture was added to a new column to separate cleaved receptor from the tag-bound cobalt resin and washed with Exchange Buffer to collect the flow through. Flow through was concentrated to ~1 ml before quantifying.

### Nuclear magnetic resonance (NMR)
Purified protein samples concentrated to ~350 μl with 10% $D_2O$ by volume were loaded into a 5 mm Shigemi microtube. Heteronuclear single quantum coherence (HSQC) spectra were collected on a Bruker Avance 800 MHz spectrometer equipped with a triple-resonance cryogenic probe, with experiments collected at 310 K using TopSpin 3.6. Experimental times were 27 h. Data were processed using NMRPipe 9.6[97] and visualized in XEASY 1.3.13[98]. CXCL12 used in this assay was purchased from Protein Foundry, L.L.C.

### Expression and Purification of VUN700, VUN701, VUN702 (NMR)
The sequences of VUN700, VUN701, VUN702 were codon-optimized for *E. coli* expression and ordered from GenScript. The nanobodies were cloned into a pET28a-6xHis-SUMO3 vector and expressed in BL21 DE3 *E.coli*. Cells were expressed at 37 °C in Luria-Bertani (LB) medium and induced with 1 mM IPTG at an OD600 of 0.8. Cultures continued to grow for 5 and a half hours before bacteria were pelleted by centrifugation and stored at −20 °C. Bacterial pellets were resuspended in ~20 mL of Buffer A (50 mM $Na_2PO_4$ (pH 8.0), 300 mM NaCl, 10 mM imidazole, 1 mM PMSF, and 0.1% (v/v) 2-mercaptoethanol (BME)) per pellet and lysed via sonication. Lysed cells were clarified at $18,000 \times g$, and the supernatant was discarded. Pellets were resuspended by sonication in ~20 mL of Buffer AD (6 M guanidinium, 50 mM $Na_2PO_4$ (pH 8.0), 300 mM NaCl, 10 mM imidazole) and spun down at $18,000 \times g$ for 20 min. Using an AKTA-Start system (GE Healthcare), the supernatant was loaded onto a Ni-NTA column equilibrated in Buffer AD. The column was washed with Buffer AD, and proteins were eluted using Buffer BD (6 M guanidinium, 50 mM sodium acetate (pH 4.5), 300 mM NaCl, and 10 mM imidazole). Proteins were refolded overnight via drop-wise dilution into a 10-fold greater volume of Refold Buffer (50 mM Tris (pH 7.6), 150 mM NaCl) with the addition of 30 mM cysteine, and 1 mM cystine. Refolded protein was concentrated in an Amicon Stirred Cell concentrator (Millipore Sigma) using a 10 kDa MWCO. Concentrated protein was added to 6-8 kDa dialysis tubing with the addition of ULP1 to cleave the N-terminal 6xHis-SUMO3-tag and dialyzed at 25 °C against Refold Buffer overnight. The AKTA-Start system was used to load the cleaved protein onto a Ni-NTA column equilibrated in VUN701 Buffer A (Refold Buffer + 10 mM Imidazole). The column was washed with VUN701 Buffer A, and the protein was eluted using VUN701 Buffer B (Refold Buffer + 500 mM Imidazole). VUN701 underwent four rounds of dialysis in 5 mM ammonium bicarbonate, lyophilized, and stored at −80 °C for further use. The purity and identity of nanobodies were confirmed by electrospray ionization mass spectrometry using a Thermo LTQ instrument and SDS-PAGE with Coomassie staining.

### Flow cytometry
HEK293 cells stably expressing ACKR3 were washed with PBS and lifted with Accumax (Invitrogen). 150k cells were transferred per well of conical 96 well plates (Greiner). Cells were washed with cold FACS buffer (0.5% bovine serum albumin (BSA) in PBS) and treated with corresponding ligands or vehicle (untreated) for 60 min at 37 °C with Assay media (0.5% BSA, 25 mM HEPES in DMEM). After treatments, cells were treated with ice-cold buffers and kept on ice until readout. Cells were washed once with FACS buffer, followed by 2 acidic washes (0.2 M acetic acid + 0.5 M NaCl). Cells were washed 3 times with FACS buffer and labeled with 10 μL/$10^6$ cells of PE-conjugated anti-ACKR3 antibody (11G8-PE, #FAB4227 R&D Systems) in FACS buffer for 1 h at 4 °C. Unbound antibody was then 3x washed away with FACS buffer. Surface ACKR3 was assessed by flow cytometry using a Guava easyCyte benchtop flow cytometer using GuavaSoft 4.5.50 software (Luminex). The mean fluorescence intensity (MFI) representing the amount of surface labeling for each experiment was quantified using Floreada Software (Floreada.io). The constitutive internalization was then represented by the ratio of the MFI of the treated samples to the non-treated controls, over each time point. Graph Pad Prism Version 10.2.0, multiple comparisons two-way ANOVA Tukey test ($*p < 0.05$).

### ACKR3 surface expression microscopy
HEK293 cells stably expressing SNAP-ACKR3 were seeded on a poly-L-lysine coated μSlide 18 well Glass Bottom of Ibidi (#81817). Cells were seeded at 5000 cells/well and allowed to attached for 24 h. Cells were treated with 100 nM CXCL12, VUN700, VUN701, VUN702 in DMEM with 0.5% BSA and 25 mM HEPES, or untreated for 1 h at 37 °C. Surface staining was achieved by using 5 μM cell-impermeable SNAP-Surface Alexa Fluor 594 (New England Biolabs) for 45 min at 4 °C then imaged using the Nikon Ti2 microscope at 100X objective. Images were processed by subtracting the background noise using the 'subtract background' option in ImageJ/Fiji.

### Basal motility
HeLa cervical cancer cells and ACKR3 KO HeLa cervical cancer cells were detached from a subconfluent 10 cm dish and seeded at 5000 cells/well on a μSlide 18 well Glass Bottom from Ibidi (#81817). These slides were poly-L-lysine coated prior to cell seeding. Cells were allowed to attached for 5 h, then the cells were washed with complete media and treated with 100 nM VUN700, VUN701, or untreated. After 1 h the cells were imaged using the Nikon Ti2 microscope at 10X objective within a controlled chamber of 37 °C and 5% $CO_2$, using a motorized stage to image each well every 30 min for 16 h. To analyse the motility of the cells, 40 cells were randomly chosen for each condition and were manually tracked using the ImageJ/Fiji Plugin 'Manual Tracking'. These tracks were compiled and used to obtain the cell trajectory plots through the ImageJ/Fiji plugin 'Chemotaxis and Migration tool' of Ibidi, and used to calculate the accumulated distance (total distance traveled) of each cell. The data shows the mean ± SD of each condition relative to their vehicle of three independent repeats. Significance was determined by one-way ANOVA Dunnett test or unpaired Welch's t test ($p < 0.0001$ (****)).

MDA-MB-231 metastatic breast cancer cells were detached from a subconfluent flask and plated at high density in the central imaging chambers of 'Ibidi μ-Slide Chemotaxis' and allowed to attach. The central imaging chamber was flushed twice with (-/+) 100 nM nanobody-containing media, and then reservoirs filled with the same treatment, giving a stable, uniform concentration over the cells throughout the experiment. 1 h after the commencement of treatment, cells were imaged on the Nikon Ti2 microscope at 10X objective within a controlled chamber of 37 °C and 5% $CO_2$, using a motorized stage to image each central chamber every 30 min for 16 h. The migration of 40 randomly chosen cells in each group were manually tracked using the ImageJ/Fiji Plugin 'Manual Tracking', and these tracks were compiled and analyzed for their trajectories by the plugin 'Chemotaxis and migration tool' (Ibidi). Accumulated distance (total distance traveled), Euclidean distance (straight line distance from cell starting point to end point) and velocity (accumulated distance/time) were analyzed. The data shows the mean +/- SD of each group relative to their internal control, combined from multiple independent repeats. Significance determined by one-way ANOVA Dunnett test ($p < 0.05$(*), 0.005 (**), 0.0005 (***), < 0.0001 (****)).

### Fluorescence Microscopy

HeLa parental cervical cancer cells and ACKR3 KO HeLa cervical cancer cells were detached from a subconfluent 10 cm dish and seeded at 10000 cells/well on a μSlide 18 well Glass Bottom (Ibidi #81817). These slides were poly-L-lysine coated prior to cell seeding. Cells were allowed to attached for 5 h, then treated for 24 h with 10 nM CXCL12-AF647 (Almac) and 1 μM IT1t (MedChem Express). The cells were then washed with cold PBS and fixed for 10 min with 4% PFA (Sigma). Then, the cells were washed again with PBS and a drop of mounting buffer (Ibidi) was added to each well. The cells were imaged using the Nikon Ti2 microscope at 100X oil objective. Images were analyzed with ImageJ/Fiji and only brightness and contract were adjusted. All conditions were recorded with identical microscope settings.

### Reporting summary

Further information on research design is available in the Nature Portfolio Reporting Summary linked to this article.

## Data availability

The HDX mass spectrometry data have been deposited to the ProteomeXchange Consortium via the PRIDE partner repository with the dataset identifiers PXD051149. All data is presented in the manuscript and supplementary file. Source data are provided with this paper.

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

## Acknowledgements

Research was supported by the ONCORNET 2.0 (ONCOgenic Receptor Network of Excellence and Training 2.0) PhD training program (CVPA, OO) funded by the European Commission for a Marie Sklodowska Curie Actions (H2020-MSCA grant agreement 860229) and by a National Institutes of Health Allergy and Infectious Disease grant (NIAID R37AI058072 to BFV). HDX-MS experiments were carried out using the facilities of the Montpellier Proteomics Platform (PPM, BioCampus Montpellier). CC was supported by the ZonMw Veni grant 09150162010212 and LDN by ZonMw Open Competition grant 0912001211007. CB acknowledges funding from the regional funds FEDER/Région Occitanie, MUSE, Labex EpiGenMed and the French Agence Nationale de la Recherche (project LEUKOCEPTOR ANR-21-CE44-0007-01).

## Author contributions

C.V.P.A., R.H., C.T.S., and M.J.S. designed the study and experimental approaches. C.V.P.A., O.O., R.S., T.D.L., L.D.N., C.C., N.Y., L.M., and S.J. carried out the experiments. J.P.B., V.B., J.D., and C.H. generated and provided reagents. C.H., B.F.V., S.G., C.B., M.S., R.H., C.T.S., and M.J.S. supervised the research. C.V.P.A., O.O., R.S., T.D.L., and L.D.N. prepared the figures. C.V.P.A., R.H., C.T.S., and M.J.S. wrote the manuscript with input from all co-authors. All authors read and approved the final manuscript.

## Competing interests

B.F.V. has an ownership interest in Protein Foundry, L.L.C. and XLock Biosciences, Inc. R.H. is affiliated with QVQ Holding BV. All other authors declare no competing interests.
