## [Transparent Peer Review file · Nature Communications]

Constitutive activity of an atypical chemokine receptor revealed by inverse agonistic nanobodies

Corresponding Author: Professor Martine Smit

Version 0:

Reviewer comments:

Reviewer #1

(Remarks to the Author)

The authors investigated two new nanobodies: VUN700 and VUN702. Both bind to the atypical chemokine receptor ACKR3, like the previously described nanobody VUN701. At difference to VUN701 the newly described nanobodies show inverse receptor activation All the nanobodies efficiently compete CXCL12 binding to ACKR3. Using BRET assays, constitutive binding of β -arrestin 2 is reduced upon binding of the two antibodies, a lesser effect is seen with β -arrestin 1. The authors go on and show with different elegant experimental approaches that the two nanobodies lock the receptor in an inactive conformation that is unable to activate β -arrestin 2. The nanobodies reduce the chemokinesis of a cancer cell line, MDA-MB-231.

The experiments are carefully performed and well presented. A major caveat of the study are the limited experimental models. From the data it appears that HEK293T and HEK293 show qualitatively similar, but the responses differ. What about other cell lines or primary cells like (early passage) HUVECs? Similarly, a single cell line was used to monitor changes in random migration. Can the findings be extended?

A second caveat is the lack of mechanistic insights. The tapping of receptors (ACKR3) at the plasma membrane is expected, but the postulated role of GRK dependent and independent as well as the role of arrestin in potential receptor signaling remain elusive. In the discussion the authors speculate on possible mechanisms, but do not provide evidence.

The effect of the nanobodies on cancer cell migration is borderline and the explanations are highly speculative. Any conclusions from the data are premature and should be removed.

Minor:

I could not find any statement on "constitutively active arrestin" in ref 32 (line 121).

Reviewer #2

(Remarks to the Author)

In the present manuscript, Perez Almeria and colleagues present two nanobodies (VUN700, VUN702) targeting the atypical chemokine receptor ACKR3 and compare them to a previously characterized antagonistic nanobody (VUN701). VUN700 and VUN702 seem to act as inverse agonists as they inhibit the basal β -arrestin:ACKR3 interaction. Both nanobodies are novel and provide potential interesting tools to study ACKR3.

Major concerns:

- The specificity of the nanobodies VUN700 and VUN702 for ACKR3 has been assessed by NanoBRET-based competition binding between Nluc-tagged ACKR3 and CXCL12-AF647. This assay, however, does not address and rule out that the nanobodies react with CXCR4, which is endogenously expressed in HeLa (used in Suppl Fig 1 or HEK293(T) cells elsewhere in the study).

In addition, displacement of CXCL12-AF647 from ACKR3 by the nanobodies is about 10-fold less efficient than with unlabeled CXCL12.

- The 'activity' of ACKR3 investigated in this manuscript is reduced to the receptor's ability to interact with β -arrestins. Although CXCL12 is known to recruit β -arrestins to engaged ACKR3, its scavenging function is, however, is β -arrestin-independent; i.e. CXCL12 internalization by ACKR3 also occurs in HEK293/HeLa-cells lacking β -arrestin1 and β -arrestin2. Whether and to which degree VUN700 and VUN702 prevent CXCL12-scavenging remains unaddressed. This would be critical for understanding the mode of action of the nanobodies and for understanding the constitutive activity of this ACKR.

- Figure1: Addition of VUN700 and VUN702 clearly decreased the pre-association of β -arrestin2 with ACKR3 in HEK293T

cells (panel A) but barely in parental HEK293 cells (panel C). These data are contradicting and the reason is unclear. Are endogenous CXCR4 levels in these two HEK293 cell line different and through heterodimerization with ACKR3 or through cross-reactivity of the nanobodies with CXCR4 jeopardize the inverse agonist effects of the nanobodies?

Moreover, the title of the legend for Figure 1 is misleading, as the nanobodies do not suppress the basal β -arrestin2 recruitment, but rather the basal interaction between the two proteins.

- GRK (particularly GRK2 and GRK5) mediated ACKR3 phosphorylation is critical for CXCL12 driven β -arrestin recruitment (ref 4, reproduced here in Fig 1D). But, CXCL12 scavenging by ACKR3 still occurs in β -arrestin-deficient cells (ref 4). By contrast, CXCL12 scavenging requires ACKR3 phosphorylation by GRKs (ref 4). Hence, that VUN700 and VUN702 have no effect on β -arrestin:ACKR3 interaction in GRK-deficient cells (Figure 1D) is obvious and does not help to characterize the nanobodies.

- Lane 198ff. The statement is wrong. Saaber and colleagues (Ref 4) have demonstrated that ACKR3 phosphorylation is essential for receptor trafficking and CXCL12 scavenging in vivo. Reference 5 does not show receptor phosphorylation, but uses mutants where potential phosphorylation sites of ACKR3 have been modified.

- Figure 6: Breast cancer metastasis is controlled by CXCL12: CXCR4 (PMID: 11242036). The molecular mechanism how VUN700 affects the basal motility of MDA-MB-231 cells in vitro remains unclear. Even less clear is how random motility would affect metastasis formation in vivo. Why do VUN700 and VUN702 have different effects on the Euclidian distance covered by migrating MDA-MB-231 cells? Does treatment with VUN700 and VUN702 affect F-actin polymerization/cytoskeletal rearrangement in MDA-MB-231 cells? What is the underlying mechanism of reduced motility as G-protein coupling is not involved? Does the administration of VUN700 and VUN702 prevent MDA-MB-231 metastasis in vivo?

Minor points:

-lane 54: whether adrenomedullin is scavenged by ACKR3 is debated (references 10-11 versus PMID: 37252916).

Reviewer #3

(Remarks to the Author)

In the current manuscript, Perez Almeria CV et al use a set of fluorescent assays in cells and structure-based methods with purified receptor to provide mechanistic insights into the regulation of ACKR3 by nanobodies. Using nanobodies that lock the receptor in a defined conformation, they establish that receptor trafficking and basal activity is dictated by its conformation, ultimately contributing to the regulation of cell motility. They use state-of-the-art techniques and for majority of experiments obtain strong results. The presented data are of interest for people working on GPCR signalling and more generally to researchers studying transmembrane receptor regulatory mechanisms.

The fluorescent-based experiments show highly significant differences between natural ligand stimulation by CXCL12 and nanobody-triggered effect, highlighting the importance of signalling events downstream of ACKR3.

The structural model based on Hydrogen/Deuterium exchange mass spectrometry analysis does need additional experimental and analytical descriptions to strengthen and validate the conclusions.

General comments:

Could difference in ACKR3 affinity, or binding kinetics, between the three nanobodies explain the differences seen in functional assays

Microscope images of cells upon trafficking of ACKR3 supporting the graphs presented would support the results shown. Results of motility assays should be tested on a ACKR3-KO cell line to validate the importance of this receptor on motility in the used assay, and ascertain that the nanobodies do not target another receptor (test on ACKR3-KO cells +/- VUNs)

Comments on the HDX-MS part:

For the HDX-MS analysis, the authors identify, in addition to the extracellular CXCL12 binding site, that intracellular conformational changes are happening in regions 248-257 and 315-320 upon binding of VUN700 and VUN702. Such intracellular conformational changes are not observed upon VUN701 binding. The authors claim that these conformational changes are representative of an inactive-like conformation triggered by inverse agonist nanobodies. Although this observation is very interesting, additional methodological and analytical details are necessary to validate the finding (please see detailed comments below).

Although representative peptides and structures are shown in Figure 3 and Supplementary Fig. 5, the amount of information available to verify data acquisition and interpretation is not sufficient and does not meet the standards defined in the following reference (Nat Methods 2019, PMID: 31249422).

In figure 3B, uptake plots for the same peptide, in three different conditions, are presented. It is very surprising to see that uptake for the apo receptor is very different between the VUN700/ VUN702 and the VUN700 condition? Why are there only technical replicates represented and not pooled as biological replicates? The data regarding the same peptide shall be presented on a single uptake graph, combining the four conditions (apo and VUN-bound) to allow proper comparison.

In Supplementary Fig. 5, uptake plot for the N-terminal peptide 27-33 shows a very unusual deuterium incorporation trend, with very short incubation showing the largest uptake. It will be critical that the authors define what is causing this extraordinary behaviour that looks almost impossible. Because this condition is the one claimed to show protection, the conclusion of nanobody binding to this peptide could be different if the results appear to be due to a technical error.

The raw files must be made available to a public database, for example PRIDE.

There is a need to present a table with all HDX-MS raw data including:

- List of peptides with details of i) boundaries, ii) sequence, iii) deuterium uptake at every timepoint and in every condition (presented both as % deuteriation and number of deuterium), iv) Retention Time, v) difference in deuterium incorporation for

every condition expressed as both %deuteration and Nb of Deuterons.

The criteria defined to set a difference to be significant must be clearly indicated (delta%, delta Nb of Deuterons and p-value)

A figure showing peptide coverage is required

Reviewer #4

(Remarks to the Author)

In this manuscript the authors present a detailed mechanistic investigation of the constitutive signaling activity of ACKR3. The new approach described derives from the application of newly reported nanobodies that bind to the extracellular face of ACKR3. The work shows that one such nanobody acts mostly as a neutral antagonist while the other two act as inverse agonists of constitutive ACKR3 beta-arrestin recruitment and internalization. The authors use this new tool to show that the constitutive activity of ACKR3 is important in the migration of a cancer cell line, a finding that would not have been possible without the new tools described here.

Overall, the manuscript is high quality, with carefully performed experiments using a variety of appropriate and state of the art techniques. The findings are important both for the field of chemokine signaling and for the application of nanobodies to modulate the function of GPCRs. This work should be suitable for publication in Nature Communications provided a few issues can be addressed.

Major comments:

*It is now relatively widely appreciated that ACKR3 can bind and respond to a variety of peptide ligands (with probably more still to be characterized). In studies of the role of ACKR3 signaling in MDA-MB-231 cells the experimental conditions include media containing 10% FBS. The authors conclude that the activity of inverse agonist nanobodies in this assay (in the absence of added chemokines) indicates that the inverse agonist (and not the antagonist) activity of the nanobodies is responsible for the biological effect. Isn't it possible that there could be peptides present in FBS that could induce activate ACKR3 and that the nanobodies are also acting as antagonists? I see that there is some difference between VUN700 and VUN701, but these differences are smaller than those between nanobodies and control conditions. Perhaps the authors could test 10% FBS in a simpler model (HEK293) to see if there is any role of nanobody antagonism in this context.

*It looks like the authors don't include the sequences of VUN700, 701, or 702 in the publication. I suggest that in order for other authors to be able to apply these tools, these sequences must be made available as soon as possible. If there are intellectual property considerations blocking this, then the relevant information (patent disclosure information and date) should be included.

*Flow cytometry experiments with HEK293 cells were performed by first lifting cells from the plate prior to treatment and staining. HEK293 cells are naturally adherent and may undergo changes in membrane protein trafficking upon detachment from the cell surface, which may result in unusual behavior. The changes in receptor levels on the cell surface observed upon nanobody treatment should be repeated with HEK293 cells adhered to the tissue culture plate surface. I don't think that all analyses need to be repeated, but there should at least be a small cross check to ensure that lifting the cells from the plate surface doesn't lead to unusual behavior.

*Other recent efforts to impart inverse agonism on constitutively active GPCRs has noted the relevance of the persistence of inverse agonist activity upon ligand washout (<https://doi.org/10.1021/jacs.3c09694>). It would be informative for the authors to try a similar experiment to see if inverse agonism persists following ligand washout.

Minor comments:

*On line 90 the text says "Nanobodies display high affinity and specificity for their target and tend to interact with non-linear, 3-dimensional shapes". A reference should be added to support this statement.

*Supporting table 1 contains EC50 and IC50 values for compounds that act either as agonists or inverse agonists. There needs to be some notation to distinguish agonists vs inverse agonists in the context of this table (or perhaps the table needs to be split into two tables)

*There is a suggestion from this data that the conformational change in beta-arrestin induced by constitutively active ACKR3 is different than the conformational change induced by ligand-activated receptor. Is this a new observation or are there literature precedents for this difference? If a new finding, this should be mentioned in the manuscript text. If previously observed in other systems this should be cited.

*For differential HDX analyses, it seems that in many cases the differences in exchange are observed at shorter time points but disappear after more prolonged incubation periods. Could the authors add a brief note on why this trend might be observed?

*In Figure 4, there seems to be a small discrepancy in the trends seen in panels a-b and panel c. VUN701 seems to increase receptor levels on the cell surface (panels a-b) but doesn't decrease levels in the endosome (panel c). Is there an explanation the authors could include on this subject in the text?

*On line 357 describe a HeLa cell line described "previously". A citation should be added here.

*The "Cell culture and transfection" section names two of the authors who contributed cell lines. This is not necessary since they are already authors on the paper.

*On line 530 the authors use the word "expect" but I think they mean "except"

*For the ACKR3 purification methods section, the authors should describe what kind of cells were used for protein production (I guess it is E coli?)

*The authors should cite a recent review that comprehensively lists nanobodies that bind to GPCRs (DOI: <https://doi.org/10.1124/molpharm.124.000974>) as this would be a valuable reference for readers of this manuscript

Reviewer #5

(Remarks to the Author)

The authors report novel nanobodies against atypical chemokine receptor 3, which stabilize the inactive state of the receptor. The two new nanobodies have distinct activities from a previously described one and seem to act as inverse agonists. The nanobodies were profiled in different assays and are shown to exhibit decreased arrestin engagement and inhibition of internalization. With biophysical methods, the conformational states of the receptor are interrogated when bound to the different nanobodies, and the data are compatible with different conformations induced by the two classes of nanobodies.

My review will focus on the biophysical characterization of the receptor based on HDX-MS and NMR, because this is my field of expertise.

The HDX-MS data convincingly reveals that nanobodies VUN700/2 allosterically induce a different state than VUN701, as evidenced by different protection factors at the intracellular side of the receptor. This is nicely in line with the biochemical data shown in the manuscript.

Here there are only minor comments:

- The error bars in figure 3 seem to be smaller than the spheres representing the data points. However, in the data shown for VUN701 the last point of the Apo curve is lower than the second to last. The authors should briefly discuss why this does not lead to re-evaluation of the error of the measurement.
- It's uncommon to write the methods section in present tense – especially since most of the rest of the methods are written in past tense.

For the NMR evaluation, the spectra clearly reveal that all nanobodies induce the inactive state of the receptor. The conclusion drawn from the differences of the peak positions in the spectra of the three nanobodies at the verge of being an overinterpretation and need to be substantiated by complementary experiments. Given the naturally relatively low resolution of the spectra, the differences in peak position are significant, if exactly identical sample preparation can be guaranteed. Just a slight change in pH can lead to stronger shifts than the differences discussed here. In my view, the data of M212 suggests three different states, whereas the data on M138 suggests that VUN700 and VUN702 induce a common state that might be slightly closer to the fully inactive state of the receptor. I believe that in conjunction with the presented HDX-MS data the interpretations are valid.

Here I also have a few comments:

Could a spectrum of the apo state be included? This would give an additional insight: the position of M138 between the nanobody and CXCL12 states should represent the activation state, which could be related to the basal activity seen in assays.

The caption seems to have been written in a rush: while the figure shows panels A-D, the caption describes panels A-C.

The signal of M138 seems to have a significantly higher resolution in the ¹³C dimension in the spectrum of VUN702 (suppl. Fig. 4). What could be the reason for this?

In summary, the HDX-MS data allows to bin the nanobodies clearly into two different classes. The NMR data doesn't add much, but showing that the receptors are all in the inactive conformation. With the knowledge of the HDX-MS data, the slight differences between the NMR spectra can be interpreted.

Version 1:

Reviewer comments:

Reviewer #1

(Remarks to the Author)

The authors have thoroughly revised the manuscript, included new data, and removed unclear sections.

Comments:

Figure 4 and 5: VUN701 "only" blocks CXCL12 binding (line 297). However, VUN701 clearly induces surface accumulation of ACKR3. The authors should explain this in more detail and exclude that accumulation at the cell surface is due to de novo synthesis. Reduced (spontaneous) internalization is less likely, because there is no accumulation at early endosomes (Fig 4C). The inclusion of protein synthesis inhibitors may also explain the late (>40 min) accumulation of ACKR3 at the cell surface in GRK ko cells stimulated with CXCL12 (Fig 5C).

Introduction line 81: please add to the list of "handful" inhibitors ref (actual 83) doi: 10.1096/fj.202002465R, a real inverse

agonist. This small molecule antagonist is being tested and is commercially available. The mechanism of action appears similar to VUN700 and VUN720 which should be added to the discussion as potential alternative to nanobody treatment.

Reviewer #2

(Remarks to the Author)

The authors have carefully revised their manuscript. Additional data improved the study. A drop of bitterness is that no functional, in vivo data have been provided to address the consequences of keeping ACKR3 in its constitutive active conformation as asked for in the first round of revision. Instead of extending their observation on dampened MDA-231 breast cancer cell motility upon treatment with the inverse agonistic nanobodies, the authors have moved that data to the supplement and down-toned the potential functional importance of a constitutive active atypical receptor. As a "tumor model" HeLa cells have been used in the revised manuscript. If in vivo application of nanobodies may is not expedient, the small molecule inverse agonists (e.g. VUF16840 or CCX771) could be used instead. In essence the revised manuscript profoundly describes the solid and careful in vitro characterization of two novel inverse agonistic nanobodies and provides new insights into the regulation of ACKR3 signaling using these nanobodies.

Minor points:

Lane 36: ..in inactive receptor conformation with decreased arrestin engagement..

Lane 96: Inhibition of receptor constitutive activity resulted in to slower cell motility.

Fig 1F is not mentioned in the result section

Lane 164: Binding of the all nanobodies protected the extracellular face from deuteration,..

Lane 238: What is the evidence for the statement that constitutive internalization of ACKR3 is independent of arrestins and GRKs?

Lane 254: reference 61 is a review article that does not show experimentally that ACKR3 contribute to cancer cell migration.

Legend to Fig 1 has not been changed as promised in the rebuttal letter.

Reviewer #3

(Remarks to the Author)

The authors have addressed the large majority of comments, but a major concern remains.

There is a point that is still unclear to me, for which the authors did not properly comment:

In the HDX-MS experiment, how can the N-terminal peptide 27-33 be more deuterated after 30 sec labelling than after 30 min and 2 hours ? This phenomenon is only seen for the apo receptor. Even if seen throughout multiple replicates, this effect cannot be justified by any rational explanation. It rather suggests that the wrong peptide was used as a reference by the software. As exclusively this condition is used to claim that the nanobodies protect/ contact this region, I would not feel comfortable in having these data published.

Reviewer #4

(Remarks to the Author)

The authors have effectively addressed my comments and concerns. The resulting manuscript will be interesting and valuable to the research community.

One minor request is listed below:

*Information on the source and product numbers for the nanobodies used in this study (from QVQ) should be listed in the methods section.

Version 2:

Reviewer comments:

Reviewer #1

(Remarks to the Author)

The authors addressed my concern about possible de novo synthesis of ACKR3 over time. The experiment included in the rebuttal is convincing. My suggestion is to present these data in the supplementary information.

No further comments.

Reviewer #3

(Remarks to the Author)

Authors have addressed my comment with a rational explanation.

It would be good to make sure that reference to the described technical issue is inserted into the manuscript.

Response to reviewers:

We would like to thank the reviewers for their thorough and critical evaluation of our manuscript. We highly appreciate their comments and suggestions, which helped to improve our manuscript. We addressed the comments point-by-point and have enclosed a revised manuscript and supplementary material documents, marked up with the changes we have made. Please find the reviewers' comments below with our answers in blue.

Reviewer #1 (Remarks to the Author)

The authors investigated two new nanobodies: VUN700 and VUN702. Both bind to the atypical chemokine receptor ACKR3, like the previously described nanobody VUN701. At difference to VUN701 the newly described nanobodies show inverse receptor activation All the nanobodies efficiently compete CXCL12 binding to ACKR3. Using BRET assays, constitutive binding of β -arrestin 2 is reduced upon binding of the two antibodies, a lesser effect is seen with β -arrestin 1. The authors go on and show with different elegant experimental approaches that the two nanobodies lock the receptor in an inactive conformation that is unable to activate β -arrestin 2. The nanobodies reduce the chemokinesis of a cancer cell line, MDA-MB-231.

The experiments are carefully performed and well presented.

A major caveat of the study are the limited experimental models. From the data it appears that HEK293T and HEK293 show qualitatively similar, but the responses differ. What about other cell lines or primary cells like (early passage) HUVECs? Similarly, a single cell line was used to monitor changes in random migration. Can the findings be extended?

We agree that the few cell types investigated presents a limitation to our current study. A fact made more apparent due to the reported differences in ACKR3 activity in different cell types can be quite drastic, including rare cases of G protein activation. Thus, we have now expanded to an additional cellular background for both the BRET assays in CHO cells and basal motility experiments in HeLa, besides other experiments performed in HEK293 and MD-MBA-231 cells. In both cases, the inverse agonistic effects were recapitulated exactly. Additionally, we had the fortuitous luck that ACKR3 knockout HeLa cells were available in time for testing for the rebuttal. These cells showed an identical decrease in migration as the WT cells treated with the inverse agonistic nanobodies. This provides stronger evidence for a role of ACKR3 in regulating the basal motility and thus we have opted to replace the original main text figure with the HeLa cell data and move the original MDA-MB-231 results to the supplementary. The results have been modified as follows:

"The decrease in ACKR3-arrestin BRET was also observed in CHO cells, confirming the effects of VUN700 and VUN702 are not limited to the HEK293 model system (Supplementary Fig. 2)."

"ACKR3 is reported to contribute to cancer cell migration, but not due to activation by CXCL12. Instead, we hypothesize that ACKR3's constitutive activity might play a role in cell motility. To resolve the influence of ACKR3 on non-chemokine driven migration, the basal or random movement of cervical cancer cells, HeLa cells, was tracked by live-single cell microscopy. HeLa cells express both ACKR3 and CXCR4 endogenously. This allows for examination of potential roles for ACKR3 in a relevant cellular context and in the presence of CXCR4. The cells showed considerable motility in the presence of 10% FBS without chemotactic stimulation (Fig. 6A). This basal motility was reduced when treated with VUN700 (Fig. 6A,B). This showed a 30% average decrease in accumulated distance traveled with inverse agonist

treatment. Inclusion of 10% FBS did not affect the inverse agonistic properties of VUN700 in the β arrestin recruitment assay (Supplementary Fig. 11). To confirm that the effects are due to ACKR3 specifically, the receptor was genomically-knocked out from these cells by CRISPR (ACKR3-KO). Without ACKR3, the cells showed reduced motility compared to WT HeLa cells, which showed no further reduction with the treatment of nanobody, thereby suggesting the effect is due specifically to ACKR3 targeting. Similar results were observed when metastatic breast cancer cells, MDA-MB-231, which express high levels of endogenous ACKR3 and CXCR4, were treated with the ACKR3 nanobodies (Supplementary Fig. 10). Basal motility of these cells was also reduced with VUN700 treatment, with a similar ~30% decrease in accumulated distance. In both cell lines VUN701 also affected basal motility (Fig. 6, supplementary Fig. 10), which may be explained by long-term treatment associated with trapping of ACKR3 on the plasma membrane, seen for all ACKR3 nanobodies in time (Fig. 4A). VUN400, a CXCR4 targeting nanobody that inhibits CXCL12 binding and CXCL12-induced chemotaxis, had no effect on basal MDA-MB-231 cell motility. These results suggest a role for the basal activity of ACKR3 in mediating non-chemotactic movement of cancer cells.”

The inclusion of another cell line (HeLa cells) and the respective ACKR3 cells was also referred to in the discussion:

“The effects of VUN700 and VUN701 on basal cancer cell (HeLa, MDA-MB-231) motility suggest that the constitutive activity of ACKR3 is implicated in migratory signaling. ...

Moreover, the basal motility in HeLa ACKR3 KO cells was severely impaired.”

A second caveat is the lack of mechanistic insights. The tapping of receptors (ACKR3) at the plasma membrane is expected, but the postulated role of GRK dependent and independent as well as the role of arrestin in potential receptor signaling remain elusive. In the discussion the authors speculate on possible mechanisms, but do not provide evidence.

While we would also prefer to build a more comprehensive analysis of all aspects of ACKR3 function, how or why the receptor performs its functions is unfortunately not straightforward even after decades of research. The trapping effects of the inverse agonists is expected, the suppression of GRK-independent internalization was not. Additional players likely contribute to the internalization (and thus scavenging) of ACKR3 and multiple efforts across several labs, including ours, are searching for these effectors.

The effect of the nanobodies on cancer cell migration is borderline and the explanations are highly speculative. Any conclusions from the data are premature and should be removed.

We agree that the effects are small, however, the effects of the nanobodies on migration are reproducible and suggest an unexplained role for ACKR3 in regulating basal motility. Moreover, the effects of ACKR3 nanobodies on basal motility were also shown in HeLa cells endogenously expressing ACKR3. Basal motility in HeLa ACKR3 KO cells was also reduced and no longer showed a difference when treated with the ACKR3 nanobodies. This new data further supports a potential role for ACKR3 in migration. We very carefully avoided making strong conclusions based on this data and have emphasized the need for follow-up studies into potential mechanisms for ACKR3 influence on migration.

Minor:

I could not find any statement on "constitutively active arrestin" in ref 32 (line 121).

As recognized by the reviewer, this exact line is not used in ref 32 (now 34) (Otun et al. PNAS 2024), but the β -arrestin1 employed in the study was C-terminally truncated to increase the

basal association of the protein with the receptor. This can be found in the material and methods of the reference under “Expression and Purification of β -Arrestin1”.

Reviewer #2 (Remarks to the Author):

In the present manuscript, Perez Almeria and colleagues present two nanobodies (VUN700, VUN702) targeting the atypical chemokine receptor ACKR3 and compare them to a previously characterized antagonistic nanobody (VUN701). VUN700 and VUN702 seem to act as inverse agonists as they inhibit the basal β -arrestin:ACKR3 interaction. Both nanobodies are novel and provide potential interesting tools to study ACKR3.

Major concerns:

- The specificity of the nanobodies VUN700 and VUN702 for ACKR3 has been assessed by NanoBRET-based competition binding between Nluc-tagged ACKR3 and CXCL12-AF647. This assay, however, does not address and rule out that the nanobodies react with CXCR4, which is endogenously expressed in HeLa (used in Suppl Fig 1 or HEK293(T) cells elsewhere in the study).

We thank the reviewer for the careful reading of the manuscript and comments. This is an important point and if the nanobodies were to be cross-reactive with CXCR4 it would complicate interpretation of the data. We have confirmed that the nanobodies compete with fluorescent CXCL12 for N-terminally Nluc-tagged CXCR4 in whole HeLa cells (an identical setup in Supplementary Fig. 1). No ACKR3-directed nanobodies decreased CXCL12 binding to the canonical receptor, indicating that the nanobodies are not cross-reactive between the sibling receptors. This data has been added to Supplementary Fig. 1 and the following line to the results.

“All three ACKR3 nanobodies bind the receptor extracellularly, compete with CXCL12, and do not show cross reactivity with CXCR4”

In addition, displacement of CXCL12-AF647 from ACKR3 by the nanobodies is about 10-fold less efficient than with unlabeled CXCL12.

The reviewer is correct that the nanobodies are more efficiently displaced by CXCL12-A647 compared to unlabeled CXCL12 (Supplementary Fig. 1). However, the core concept of this study is the suppression of the constitutive activity of ACKR3 and not the use of the nanobodies to compete with CXCL12.

- The ‘activity’ of ACKR3 investigated in this manuscript is reduced to the receptor’s ability to interact with β -arrestins. Although CXCL12 is known to recruit β -arrestins to engaged ACKR3, its scavenging function is, however, β -arrestin-independent; i.e. CXCL12 internalization by ACKR3 also occurs in HEK293/HeLa-cells lacking β -arrestin1 and β -arrestin2. Whether and to which degree VUN700 and VUN702 prevent CXCL12-scavenging remains unaddressed. This would be critical for understanding the mode of action of the nanobodies and for understanding the constitutive activity of this ACKR.

Given that the nanobodies compete for binding with CXCL12, they would also inhibit scavenging purely by acting as competitive antagonists (Supplementary Fig. 1). Due to this relationship, the only outcome from CXCL12 scavenging will be competitive inhibition. We

present the effects of the nanobodies on constitutive internalization, a key component for CXCL12 scavenging (Saaber et al. Cell Reports 2019, Schafer et al. Mol Pharm 2023), and observe inhibition of both GRK-dependent and independent internalization mechanisms. If simultaneous binding of the nanobodies and CXCL12 to ACKR3 were possible, the suppression of receptor trafficking would also impair scavenging. The following text has been added to the discussion to note this impossibility.

“Thus, even if co-binding with CXCL12 were possible, the nanobodies would prevent chemokine scavenging by inhibiting internalization of the receptor.”

- Figure1: Addition of VUN700 and VUN702 clearly decreased the pre-association of β -arrestin2 with ACKR3 in HEK293T cells (panel A) but barely in parental HEK293 cells (panel C). These data are contradicting and the reason is unclear. Are endogenous CXCR4 levels in these two HEK293 cell line different and through heterodimerization with ACKR3 or through cross-reactivity of the nanobodies with CXCR4 jeopardize the inverse agonist effects of the nanobodies?

The reviewer is correct that these different cell lines behave differently in terms of magnitude of the BRET changes. Importantly, treatment with inverse agonistic nanobodies led to a decrease in arrestin interaction with ACKR3 in HEK293 cells just as in HEK293T cells. As noted above, there is no detectable cross-reactivity of the nanobodies with CXCR4, which argues against that being a potential issue. The more likely explanation is a difference in levels of GRK expression and basal activity. It has been previously shown by ourselves and others (Schafer et al. Mol Pharm 2023, Sarma et al. Nat Comm 2023) that phosphorylation of stimulated ACKR3 is dominated by GRK5. GRK2 can also contribute under certain conditions, such as with co-activation of CXCR4 (Schafer et al. Mol Pharm 2023) or with over expression of the kinase (Saaber et al. Cell Reports 2019). In HEK293T cells, the contribution of GRK2 to ACKR3 function is greater than was observed in the HEK293A cells used in other studies (Saaber et al. Cell Reports 2019, Zarca et al. Cells 2021). Whether this is the result of greater GRK2 expression or increased basal canonical GPCR activity is unclear and either scenario leads to greater GRK2 activity even on basal receptors therefore altering the initial constitutive association and the degree which it can be reduced. This is also reflected by a 50% increase in the window for the CXCL12 treated condition, which likely reaches a higher fold increase due to a lower initial level.

Moreover, the title of the legend for Figure 1 is misleading, as the nanobodies do not suppress the basal β -arrestin2 recruitment, but rather the basal interaction between the two proteins.

We thank the reviewer for the suggestion and have changed the figure title to ‘interaction’.

- GRK (particularly GRK2 and GRK5) mediated ACKR3 phosphorylation is critical for CXCL12 driven β -arrestin recruitment (ref 4, reproduced here in Fig 1D). But, CXCL12 scavenging by ACKR3 still occurs in β -arrestin-deficient cells (ref 4). By contrast, CXCL12 scavenging requires ACKR3 phosphorylation by GRKs (ref 4). Hence, that VUN700 and VUN702 have no effect on β -arrestin:ACKR3 interaction in GRK-deficient cells (Figure 1D) is obvious and does not help to characterize the nanobodies.

We thank the reviewer for the comment, we believe characterization of the ACKR3 nanobodies in these assays is still of added value. First, while it is true that CXCL12 scavenging is primarily mediated by active internalization following GRK phosphorylation, some chemokine uptake

occurs by constitutive internalization which is GRK independent (shown in Schafer et al. Mol Pharm 2023 in Fig. 1B using GRK-deficient cells). Additionally, both Saaber et al. Cell Rep 2019 (shown in Fig. 3D) and Schafer et al. Mol Pharm 2023 (shown in Fig. 4E) show that phosphorylation deficient ACKR3 (ST/A construct) can scavenge CXCL12, albeit less than when the receptor can be phosphorylated.

Secondly, our recent HDX study (Otun et al. PNAS 2024) demonstrated that apo-ACKR3 interacts with arrestin in the absence of phosphorylation in vitro. This suggests that a basal interaction may be phosphorylation, and thus GRK, independent. Therefore, we tested whether the basal complexes in a cell system were dependent on GRKs or not using the GRK-deficient HEK cells. This was explained in detail in the original manuscript on lines 120-122 (now 126-128 in the marked revised manuscript).

- Lane 198ff. The statement is wrong. Saaber and colleagues (Ref 4) have demonstrated that ACKR3 phosphorylation is essential for receptor trafficking and CXCL12 scavenging in vivo. Reference 5 does not show receptor phosphorylation, but uses mutants where potential phosphorylation sites of ACKR3 have been modified.

While we appreciate the reviewer's perspective, we interpret the statement on Line 198 and the references Saaber et al. Cell Report 2019 (Ref4) and Schafer et al. Mol Pharm 2023 (Ref5) differently, based on our understanding of the original studies. Saaber 2019 (Ref4) showed in vivo that phosphorylation deficient ACKR3 retains the ability to internalize CXCL12, albeit less efficiently than when GRKs are able to phosphorylate the receptor (Fig. 3D, in Saaber et al. Cell Rep 2019). They also showed that GRK2 and GRK5 can phosphorylate the receptor (Fig. 1D in Saaber et al. Cell Rep 2019). In our previous work (Schafer 2023, Ref5) we demonstrated that partial CXCL12 scavenging persisted in the absence of receptor phosphorylation (Fig. 1B, 4E in Schafer et al. Mol. Pharm 2023). This was tested both by genomic knockout of the GRKs and mutation of specific phosphorylation sites that we identified by mass spectrometry following specific GRK2 or GRK5 phosphorylation (Fig. 4A, Supplemental Fig. 2 in Schafer et al. Mol. Pharm 2023). This work represents the only instance where GRK-specific phosphorylation patterns have been resolved for ACKR3. (Saaber et al. Cell Reports 2019 only characterizes a single position by phospho-specific antibody, while Zarca et al. Cells 2021 and Sarma et al. Nat. Comm. 2023 only perform mutagenesis of specific serine and threonine residues and do not test chemokine scavenging.) Additionally, we show ACKR3 shows robust constitutive internalization that is independent of GRKs (Fig. 1C in Schafer et al. Mol. Pharm 2023) and C-terminal phosphorylation (Fig. 4F in Schafer et al. Mol. Pharm 2023). It is clear from both studies that constitutive, GRK-independent internalization contributes to CXCL12 scavenging by ACKR3 and is independent of receptor phosphorylation.

- Figure 6: Breast cancer metastasis is controlled by CXCL12:CXCR4 (PMID: 11242036). The molecular mechanism how VUN700 affects the basal motility of MDA-MB-231 cells in vitro remains unclear. Even less clear is how random motility would affect metastasis formation in vivo. Why do VUN700 and VUN702 have different effects on the Euclidian distance covered by migrating MDA-MB-231 cells?

The reviewer is correct that directed breast cancer cell migration is mediated by CXCL12 activation of CXCR4. Nevertheless, treatment with the CXCR4 specific nanobody, VUN400, showed no effect on the basal motility, and therefore confirming that CXCR4 is not driving non-

chemotactic migration. Future investigations are seeking to connect these basal activities of ACKR3 with other cellular machinery, however this is beyond the scope of this study.

The inverse agonist VUN700 and neutral antagonist VUN701 were tested in the migration experiment using MDA-MB-231 cells (now Supplementary Fig. 12). In both total and Euclidian distance, VUN701 showed a lesser effect than VUN700. In HeLa cells, now presented in Fig. 6, the effects on motility of VUN700 and VUN701 treatment are very similar. We have added the following to the results highlighting the effect of the neutral antagonist on cancer cell motility:

“In both cell lines VUN701 also affected basal motility (Fig. 6, supplementary Fig. 12), which may be explained by long-term treatment associated with trapping of ACKR3 on the plasma membrane, seen for all ACKR3 nanobodies in time (Fig. 4A).”

VUN702, the other inverse agonist, was not included in these experiments.

Does treatment with VUN700 and VUN702 affect F-actin polymerization/cytoskeletal rearrangement in MDA-MB-231 cells? What is the underlying mechanism of reduced motility as G-protein coupling is not involved? Does the administration of VUN700 and VUN702 prevent MDA-MB-231 metastasis in vivo?

We agree these are great questions and are the topic of follow up investigations already underway in our group.

Minor points:

-lane 54: whether adrenomedullin is scavenged by ACKR3 is debated (references 10-11 versus PMID: 37252916).

We appreciate the reviewer pointed out the further analysis of ACKR3 and adrenomedullin interactions. The ambiguity with regards to ACKR3 scavenging of adrenomedullin has been added to the text along with the suggested reference.

“Besides chemokines, ACKR3 is activated by opioid peptides (BAM22, enkephalins, and dynorphins)^{9,10} and pro-adrenomedullin derivatives (adrenomedullin and PAMP-12)^{11,12}, although the agonist properties of adrenomedullin has recently been disputed¹³.”

Reviewer #3 (Remarks to the Author):

In the current manuscript, Perez Almeria CV et al use a set of fluorescent assays in cells and structure-based methods with purified receptor to provide mechanistic insights into the regulation of ACKR3 by nanobodies. Using nanobodies that lock the receptor in a defined conformation, they establish that receptor trafficking and basal activity is dictated by its conformation, ultimately contributing to the regulation of cell motility. They use state-of-the-art techniques and for majority of experiments obtain strong results. The presented data are of interest for people working on GPCR signalling and more generally to researchers studying transmembrane receptor regulatory mechanisms.

The fluorescent-based experiments show highly significant differences between natural ligand stimulation by CXCL12 and nanobody-triggered effect, highlighting the importance of signalling events downstream of ACKR3.

The structural model based on Hydrogen/Deuterium exchange mass spectrometry analysis does need additional experimental and analytical descriptions to strengthen and validate the conclusions.

General comments:

1. Could difference in ACKR3 affinity, or binding kinetics, between the three nanobodies explain the differences seen in functional assays

We thank the reviewer for the comments. We considered whether the differences between the nanobodies could be explained by different properties, with the antagonist VUN701 being a weaker interactor than either of the inverse agonists. However, as shown in Supplementary Fig. 1, the affinities of the nanobodies are not drastically different. Furthermore, the sequences of VUN700 and VUN701 are very similar, while the second inverse agonist VUN702 differs. Thus, the predicted structures and binding modes of VUN700 and VUN701 are expected to be similar, while VUN702 may act differently, despite producing the same downstream effect as VUN700. The differences between the inverse agonists is the subject of a follow-up publication, while this manuscript focuses on the suppression of basal ACKR3 activity.

2. Microscope images of cells upon trafficking of ACKR3 supporting the graphs presented would support the results shown.

This is a good point and we agree that imaging would bolster the trapping data derived from the flow cytometry and BRET analyses included in Fig. 4. Thus, we stained the surface localized SNAP-ACKR3 with a membrane impermeable SNAP dye following treatment with the nanobodies or CXCL12. Consistent with our flow cytometry data, after 45 min of treatment, the CXCL12-treated cells returned to levels similar to the mock treated condition. In contrast, all three nanobodies lead to a trapping of the receptor at the surface.

3. Results of motility assays should be tested on a ACKR3-KO cell line to validate the importance of this receptor on motility in the used assay, and ascertain that the nanobodies do not target another receptor (test on ACKR3-KO cells +/- VUNs)

We thank the reviewer for the suggestion. Since submission, we generated ACKR3 KO HeLa cells by CRISPR, which have allowed us to address this question. The ACKR3-KO cells showed less basal motility than WT HeLa cells. Additionally, the knockout cells showed no further reduction in cell movement with nanobody treatment. These experiments further validate the contribution of ACKR3 to these basal effects as well as demonstrating the phenomenon is not limited to the MDA-MB-231 breast cancer cells in the original manuscript. Due to the clear demonstration of ACKR3's role, we have placed this new data from HeLa cells into the main text in place of the original Fig. 6 and moved the MDA-MB-231 data to Supplementary Fig. 12.

Comments on the HDX-MS part:

For the HDX-MS analysis, the authors identify, in addition to the extracellular CXCL12 binding site, that intracellular conformational changes are happening in regions 248-257 and 315-320 upon binding of VUN700 and VUN702. Such intracellular conformational changes are not observed upon VUN701 binding. The authors claim that these conformational changes are representative of an inactive-like conformation triggered by inverse agonists nanobodies.

Although this observation is very interesting, additional methodological and analytical details are necessary to validate the finding (please see detailed comments below).

Although representative peptides and structures are shown in Figure 3 and Supplementary Fig. 5, the amount of information available to verify data acquisition and interpretation is not sufficient and does not meet the standards defined in the following reference (Nat Methods 2019, PMID: 31249422).

We thank the reviewer for pointing that out. As mentioned below, we have now clarified/added the missing data in the revised version: Supplementary figure 6 showing the sequence coverage and representative heatmap for every analysed Nb, Supplementary data 1 for HDX summary tables, and Supplementary data 2 for HDX data tables..

4. In figure 3B, uptake plots for the same peptide, in three different conditions, are presented. It is very surprising to see that uptake for the apo receptor is very different between the VUN700/ VUN702 and the VUN700 condition? Why are there only technical replicates represented and not pooled as biological replicates? The data regarding the same peptide shall be presented on a single uptake graph, combining the four conditions (apo and VUN-bound) to allow proper comparison.

The apo receptor is not similar since we did not perform all three nanobodies using the same biological preparation of ACKR3 (we are unfortunately limited in the yield of production/purification of these receptors from insect cells). The data shown for the uptake plots are representative of two separate biological replicates in which VUN700/702 are compared to a different apo reference to VUN701 (i.e. same apo VUN700/702, different apo for VUN701).

From our experience and others (see for example the peer review file of Jia et al. 2020 Nature communications, <https://doi.org/10.1038/s41467-020-20032-3>), the variability among biological replicates of membrane proteins is greater than that of soluble proteins, which is likely influenced by the presence of detergents and the variation of final detergent concentration during the multiple purification/concentration steps used for every biological preparation. Therefore, absolute exchange levels fluctuate between biological preparations, making it impossible to pool all biological replicates together. This increased variability also restricts the extent of quantitative analysis that can be conducted. Despite this, regions exhibiting significant Δ HDX remain qualitatively consistent across biological replicates and we typically perform biological replicates to make sure to identify regions that show systematically a consistent and reproducible Δ HDX trend.

We added the following statement in the results section to clarify this aspect:

“It is important to note that due to limited protein yield, the apo receptor reference was not derived from the same biological preparation for all nanobody conditions. To account for the inherent variability between membrane protein preparations, each nanobody condition was compared to its corresponding apo. Despite these variations, the qualitative trends in Δ HDX remain consistent across biological replicates.”

5. In Supplementary Fig. 5, uptake plot for the N-terminal peptide 27-33 shows a very unusual deuterium incorporation trend, with very short incubation showing the largest uptake. It will be critical that the authors define what is causing this extraordinary behaviour that look almost

impossible. Because this condition is the one claimed to show protection, the conclusion of nanobody binding to this peptide could be different if the results appear to be due to a technical error.

We were also surprised by this peculiar behaviour, and thank the reviewer for pointing this out as we realize that our explanation given in the original manuscript (lines 179-182) deserves further development.

Technical aspects: The behaviour at the N-terminal region of ACKR3 was reproducible in all 9 datasets we performed (3 biological replicates done in technical triplicates for every analysed nanobody). For every biological dataset, we acquire the different deuteration timepoints in a random order to ensure that observed uptake reflects changes in the conformational dynamics of intact proteins rather than potential effects of protein integrity. In each dataset, we observed the same trend at the level of residues 25-33 (spanned by multiple peptides), i.e. a high protection in the presence of the nanobodies only at the shortest deuteration timepoints. This N-terminal region is normally non-structured in the apo form, which is visible since we already reach maximum observed deuteration starting 30 sec (uptake plots Fig. 1B below) and we do not have any increased uptake (even after 12h deuteration). Below a representative stacked uptake plot for peptide 27-33 (Fig. 1A) as well as uptake plots of neighbouring peptides from the same dataset (Fig. 1B), in which we have performed an additional 2 min deuteration timepoint; these results clearly show that the protection at 30 sec and 2 min is not a technical error.

Figure 1: A) Stacked spectral plots of a representative 27-33 peptide (ACKR3 apo versus ACKR3 + VUN702 Nanobody) showing all assigned peaks for the apo state and the Nanobody-bound state in triplicates for every deuteration timepoints (from bottom to top: 0, 30 sec, 2 min, 5 min, 30 min and 120 min). B) Uptake plots from the same dataset showing neighboring peptides 17-25, 17-26 and 17-22 at maximum observed deuteration starting 30 sec, as opposed to peptides 23-30, 26-31 and 27-33 all showing increased protection in the presence of the nanobody only at the shortest deuteration timepoints. C) Structure of the

ACKR3/CXCL12 complex (PDB ID 7SK3) showing the induced β -strand at ACKR3 N-terminus (residues 30-33).

Structural interpretation in light of the conformational dynamics of ACKR3 N-terminus: the structure of ACKR3 bound to its chemokine ligand CXCL12 revealed significant differences compared to the structures of classical chemokine receptors, notably at the level of receptor's N-terminus. More specifically, CXCL12 is rotated by 70–80° in ACKR3 relative to its orientation in CXCR4 (Liu et al., 2024; Yen et al., 2022) leading to the formation of a β -strand at the N-terminus of ACKR3 to contribute to the CXCL12 β -core (Fig. 1C). This feature has not been observed in other chemokine receptors.

In our current study, the N-terminal region of ACKR3 that shows this “peculiar” behavior in the presence of the analyzed nanobodies correlates to the N-terminus of ACKR3 that becomes structured in the presence of the chemokine ligand CXCL12. Therefore, we suspect that the nanobodies lead to the structuring of this N-terminal part in a highly dynamic manner, which explains why we observe a protection in the presence of the nanobodies only at the shortest deuteration times (30 sec, 2 min), before reaching a maximum deuteration since the area is highly dynamic.

We added the following statement in the results section to clarify this aspect:

“More specifically, the N-terminal region spanning residues 25-33 exhibited reproducible and significant protection in the presence of nanobodies, but only at short deuteration time points. This suggests that the nanobodies induce a local dynamic structuring in this region, which remains unstructured in the apo form.”

6. The raw files must be made available to a public database, for example PRIDE.

As mentioned in the last section of the manuscript “Data availability” of the Methods part (lines 651-653), all the HDX raw data files have been deposited to the ProteomeXchange Consortium via the PRIDE partner repository with the dataset identifiers PXD051149. This will be made public upon manuscript publication. The reviewer can access all the data using the Reviewer account detail given during manuscript submission.

There is a need to present a table with all HDX-MS raw data including:

7.- List of peptides with details of i) boundaries, ii) sequence, iii) deuterium uptake at every timepoint and in every condition (presented both as % deuteration and number of deuterium), iv) Retention Time, v) difference in deuterium incorporation for every condition expressed as both %deuteration and Nb of Deuterons.

Two supplementary xls data sets are now added in the supplementary information in respect with the general guidelines defined by Nat Methods 2019, PMID: 31249422.

-HDX summary tables (Supplementary data 1)

-HDX data tables (Supplementary data 2)

8. The criteria defined to set a difference to be significant must be clearly indicated (delta%, delta Nb of Deuterons and p-value)

Statistical analysis of the data sets was performed using the Deuterios 2.0 software (Lau et al. 2021 *Bioinformatics*, doi: 10.1093/bioinformatics/btaa677) by applying a peptide-level significance test with 99% confidence interval (B-H adjusted p-value<0.01) to analyse each dataset (triplicate technical points performed for each biological replicate). All supporting data and meta-data, showing the delta% and delta Nb of Deuterons, are reported in the Supplementary xls data files (“HDX summary tables” and “HDX data tables”) that are now added.

This is now added in the methods section.

9. A figure showing peptide coverage is required.

We have now added one supplementary figure (Supplementary Fig. 6) for a representative peptide coverage map obtained and corresponding heatmap for every analysed nanobody. HDX summary table (Supplementary data 1) showing HDX reaction details, time course, total number of peptides, sequence coverage, average peptide length, redundancy, replicates and average SD for all 9 experimental datasets are now added in the supplementary xls datasets.

Reviewer #4 (Remarks to the Author):

In this manuscript the authors present a detailed mechanistic investigation of the constitutive signaling activity of ACKR3. The new approach described derives from the application of newly reported nanobodies that bind to the extracellular face of ACKR3. The work shows that one such nanobody acts mostly as a neutral antagonist while the other two act as inverse agonists of constitutive ACKR3 beta-arrestin recruitment and internalization. The authors use this new tool to show that the constitutive activity of ACKR3 is important in the migration of a cancer cell line, a finding that would not have been possible without the new tools described here.

Overall, the manuscript is high quality, with carefully performed experiments using a variety of appropriate and state of the art techniques. The findings are important both for the field of chemokine signaling and for the application of nanobodies to modulate the function of GPCRs. This work should be suitable for publication in *Nature Communications* provided a few issues can be addressed.

Major comments:

*It is now relatively widely appreciated that ACKR3 can bind and respond to a variety of peptide ligands (with probably more still to be characterized). In studies of the role of ACKR3 signaling in MDA-MB-231 cells the experimental conditions include media containing 10% FBS. The authors conclude that the activity of inverse agonist nanobodies in this assay (in the absence of added chemokines) indicates that the inverse agonist (and not the antagonist) activity of the nanobodies is responsible for the biological effect. Isn't it possible that there could be peptides present in FBS that could induce activate ACKR3 and that the nanobodies are also acting as antagonists? I see that there is some difference between VUN700 and VUN701, but these differences are smaller than those between nanobodies and control conditions. Perhaps

the authors could test 10% FBS in a simpler model (HEK293) to see if there is any role of nanobody antagonism in this context.

We agree that the FBS included in the migration experiment may complicate observing ACKR3's specific role. FBS was included in this experiment to reflect a more physiologically relevant condition and was associated with an increased basal cellular motility. To determine whether FBS may impact the observed inverse agonism, β -arrestin recruitment to ACKR3 was measured in HEK293T cells by BRET in 10% FBS conditions. These experiments revealed identical results as the non-FBS experiments and have been included in the manuscript in Supplementary Fig. 10 and referred to in the results section discussing Figure 6.

*It looks like the authors don't include the sequences of VUN700, 701, or 702 in the publication. I suggest that in order for other authors to be able to apply these tools, these sequences must be made available as soon as possible. If there are intellectual property considerations blocking this, then the relevant information (patent disclosure information and date) should be included.

All nanobodies used in this study were supplied by QVQ. We have indicated this in the material section now. All these molecules are therefore publicly available to others for research purposes (product numbers Q84, Q123, Q125 and Q126).

*Flow cytometry experiments with HEK293 cells were performed by first lifting cells from the plate prior to treatment and staining. HEK293 cells are naturally adherent and may undergo changes in membrane protein trafficking upon detachment from the cell surface, which may result in unusual behavior. The changes in receptor levels on the cell surface observed upon nanobody treatment should be repeated with HEK293 cells adhered to the tissue culture plate surface. I don't think that all analyses need to be repeated, but there should at least be a small cross check to ensure that lifting the cells from the plate surface doesn't lead to unusual behavior.

We thank the reviewer for raising this interesting point. To confirm that the trapping effect of the nanobodies was not an artifact of the detached system, we repeated the experiment with cells that were first treated then lifted, stained, and measured by flow cytometry. The effects were identical to the lifted condition, see the figure below.

*Other recent efforts to impart inverse agonism on constitutively active GPCRs has noted the relevance of the persistence of inverse agonist activity upon ligand washout (<https://doi.org/10.1021/jacs.3c09694>). It would be informative for the authors to try a similar experiment to see if inverse agonism persists following ligand washout.

This is an interesting point to consider and we would expect a similar phenomenon to occur here. We have tested this washout experiment and observed differences in the off-rates of the inverse agonistic nanobodies, which warrant further elucidation. As this manuscript is focused on the constitutive activity of ACKR3, we will be including these new observations in a future publication delineating the differences between these inverse agonists.

Minor comments:

*On line 90 the text says "Nanobodies display high affinity and specificity for their target and tend to interact with non-linear, 3-dimensional shapes". A reference should be added to support this statement.

The following citations have been added to support this statement: De Geest et al. PNAS 2006, Muyldermans and Wyns Trends Biochem Sci 2001.

*Supporting table 1 contains EC50 and IC50 values for compounds that act either as agonists or inverse agonists. There needs to be some notation to distinguish agonists vs inverse agonists in the context of this table (or perhaps the table needs to be split into two tables)

Labels have been added to Supplementary Table 1 to clarify the function of the compounds tested.

*There is a suggestion from this data that the conformational change in beta-arrestin induced by constitutively active ACKR3 is different than the conformational change induced by ligand-activated receptor. Is this a new observation or are there literature precedents for this difference? If a new finding, this should be mentioned in the manuscript text. If previously observed in other systems this should be cited.

We thank the reviewer for noting the implied conclusion that extends from our tests with the arrestin-FLaSH sensors. Specifically, that these results imply distinct conformations for the constitutive activity compared to the agonist-promoted activation. We previously showed similar differences in liganded vs non-liganded active forms also exist for the constitutively active viral GPCR US28. A comparison to and citation of this work has been added to the results section.

"This implies that either the basal interaction of ACKR3 and β -arrestin2 does not induce a conformational change in the arrestins and likewise suggests the constitutively active conformation of ACKR3 is different than that promoted by agonist stimulation. Similarly distinct active receptor conformations were observed for the viral GPCR US28 (De Groof et al. Nat Comm 2021)."

*For differential HDX analyses, it seems that in many cases the differences in exchange are observed at shorter time points but disappear after more prolonged incubation periods. Could the authors add a brief note on why this trend might be observed?

This trend was only visible at the level of peptides spanning the unstructured N-terminal region that appear to become structure upon binding to the nanobodies (see our answer to comment N°5 from Reviewer 3 above).

*In Figure 4, there seems to be a small discrepancy in the trends seen in panels a-b and panel c. VUN701 seems to increase receptor levels on the cell surface (panels a-b) but doesn't decrease levels in the endosome (panel c). Is there an explanation the authors could include on this subject in the text?

We note that there appears a discrepancy between plasma membrane trapping by VUN701 and an apparent lack of change at early endosomes. These data appear to suggest that VUN701 alters the trafficking of ACKR3 differently than the inverse agonists and that receptor conformation determines where a receptor might be relocated. We have added the following explanation to the results:

“The robust ACKR3 trapping at the plasma membrane, without depletion from early endosomes, of VUN701, suggests that ACKR3 is trafficked differently depending on receptor conformation.”

*On line 357 describe a HeLa cell line described "previously". A citation should be added here.

We regret the omission and have fixed the origin of the knock-in HeLa cell lines.

“NanoLuc-CXCR4 and NanoLuc-ACKR3 CRISPR Knock In HeLa cells were a kind gift from Stephen Hill from University of Nottingham⁸².”

*The "Cell culture and transfection" section names two of the authors who contributed cell lines. This is not necessary since they are already authors on the paper.

We have corrected this inclusion and the line thanking our co-authors has been removed.

*On line 530 the authors use the word "expect" but I think they mean "except"

Thank you for noting the error, this has been fixed.

*For the ACKR3 purification methods section, the authors should describe what kind of cells were used for protein production (I guess it is E coli?)

We regret the omission especially due to different purification methods employed here. In both methodologies, ACKR3 was expressed and purified from Sf9 cells. This has been added to the methods section clearly for both purification protocols.

*The authors should cite a recent review that comprehensively lists nanobodies that bind to GPCRs (DOI: <https://doi.org/10.1124/molpharm.124.000974>) as this would be a valuable reference for readers of this manuscript.

We have added this excellent review to our list of citations for nanobodies applied to GPCRs.

Reviewer #5 (Remarks to the Author):

The authors report novel nanobodies against atypical chemokine receptor 3, which stabilize the inactive state of the receptor. The two new nanobodies have distinct activities from a previously described one and seem to act as inverse agonists. The nanobodies were profiled in different assays and are shown to exhibit decreased arrestin engagement and inhibition of internalization. With biophysical methods, the conformational states of the receptor are

interrogated when bound to the different nanobodies, and the data are compatible with different conformations induced by the two classes of nanobodies.

My review will focus on the biophysical characterization of the receptor based on HDX-MS and NMR, because this is my field of expertise.

The HDX-MS data convincingly reveals that nanobodies VUN700/2 allosterically induce a different state than VUN701, as evidenced by different protection factors at the intracellular side of the receptor. This is nicely in line with the biochemical data shown in the manuscript.

The HDX-MS data convincingly reveals that nanobodies VUN700/2 allosterically induce a different state than VUN701, as evidenced by different protection factors at the intracellular side of the receptor. This is nicely in line with the biochemical data shown in the manuscript.

Here there are only minor comments:

- The error bars in figure 3 seem to be smaller than the spheres representing the data points. However, in the data shown for VUN701 the last point of the Apo curve is lower than the second to last. The authors should briefly discuss why this does not lead to re-evaluation of the error of the measurement.

The error bars reflect the SD from the 3 technical replicates performed for one single biological replicate. Thus, they are representative of the repeatability of the uptake within this biological replicate. The lower uptake at the last timepoint is likely due to a D/H back-exchange in this chosen dataset. In the revised version, we chose to represent the uptake plot from another dataset (dataset 1 instead of the originally chosen dataset 3, see figure below) for VUN701 that better reflects the fact that we do not see any reproducible statistically significant difference for peptide 315-320 in the presence of VUN701.

- It's uncommon to write the methods section in present tense – especially since most of the rest of the methods are written in past tense.

This is now corrected in the revised version.

For the NMR evaluation, the spectra clearly reveal that all nanobodies induce the inactive state of the receptor. The conclusion drawn from the differences of the peak positions in the spectra of the three nanobodies at the verge of being an overinterpretation and need to be substantiated by complementary experiments. Given the naturally relatively low resolution of the spectra, the differences in peak position are significant, if exactly identical sample preparation can be guaranteed. Just a slight change in pH can lead to stronger shifts than the

differences discussed here. In my view, the data of M212 suggests three different states, whereas the data on M138 suggests that VUN700 and VUN702 induce a common state that might be slightly closer to the fully inactive state of the receptor. I believe that in conjunction with the presented HDX-MS data the interpretations are valid.

Here I also have a few comments:

Could a spectrum of the apo state be included? This would give an additional insight: the position of M138 between the nanobody and CXCL12 states should represent the activation state, which could be related to the basal activity seen in assays.

While we do have a spectra of the Apo ACKR3 (shown below), we have not included this in the manuscript for a variety of reasons. The apo spectrum lacks M212 and has a more inverse agonist like M138 position. In these spectra, this difference does not seem to be due to the difference in dynamics, but rather is a reflection of the presence of ligand. We believe the better comparison (and less confusing for a general audience) is only using the nanobodies which all bind in a similar way with nearly identical sequences.

The caption seems to have been written in a rush: while the figure shows panels A-D, the caption describes panels A-C.

Thank you for pointing this out to us. We have corrected the figure legend as follows:

Figure 2. NMR-based structural characterization of ACKR3 upon nanobodies VUN700 binding reveals a relatively more pronounced "OFF" state of ACKR3 than VUN701-bound state. **A)** ACKR3 structure (7SK6 PDB)12 with NMR peaks M1383x46 and M2125x39 depicted. **B)** ^1H - ^{13}C HSQC spectra of ^{13}C -e-methionine labeled ACKR3 bound to the unlabeled agonist CXCL12 (blue), antagonist nanobody VUN701 (purple), or either inverse agonist nanobody VUN700 or VUN702 (green and yellow, respectively). **C)** Overlay of M2125x39 peaks from all ACKR3 complexes. **D)** Overlay of M1383x46 peaks from all ACKR3 complexes. The upfield peak positions (^1H : ~ 1.3 ppm) of M1383x46 among agonist-bound states supports ring-current shifts due to aromatic side chain interactions.

The signal of M138 seems to have a significantly higher resolution in the ^{13}C dimension in the spectrum of VUN702 (suppl. Fig. 4). What could be the reason for this?

This is an interesting observation. First and foremost, the difference in these spectra are not due to concentration. The ACKR3-VUN700 spectra was collected at $57\ \mu\text{M}$ and the ACKR3-VUN702 spectrum was collected at $52\ \mu\text{M}$. We believe this difference is due to a small improvement in stabilization of the ACKR3 structure around M138 when VUN702 is bound. Without reaching statistical significance on these assays or in these NMR spectra, we have left this out of our analysis.

In summary, the HDX-MS data allows to bin the nanobodies clearly into two different classes. The NMR data doesn't add much, but showing that the receptors are all in the inactive conformation. With the knowledge of the HDX-MS data, the slight differences between the NMR spectra can be interpreted.

Thank you for this great feedback. We also believe that this NMR data should not be taken alone, but rather with the HDX data to fully gain mechanistic insight into ACKR3 inverse agonism. We appreciate your expertise and comments.

Reviewer #1 (Remarks to the Author):

The authors have thoroughly revised the manuscript, included new data, and removed unclear sections.

Comments:

Figure 4 and 5: VUN701 "only" blocks CXCL12 binding (line 297). However, VUN701 clearly induces surface accumulation of ACKR3. The authors should explain this in more detail and exclude that accumulation at the cell surface is due to de novo synthesis. Reduced (spontaneous) internalization is less likely, because there is no accumulation at early endosomes (Fig 4C). The inclusion of protein synthesis inhibitors may also explain the late (>40 min) accumulation of ACKR3 at the cell surface in GRK ko cells stimulated with CXCL12 (Fig 5C).

We thank the reviewer for raising this concern. The core of our interpretation is that the nanobodies trap the receptors on the surface by inhibiting constitutive internalization mechanisms. While this does not explicitly exclude the possibilities of the accumulated receptors coming from de novo protein synthesis, as these proteins would be trapped just the same as those constitutively returning to the PM following internalization. We suspect this is not the case for the following reasons. Firstly, the constitutive internalization of ACKR3 is well documented (<https://doi.org/10.1038/onc.2010.212>, <https://doi.org/10.1074/jbc.m111.335679>, <https://doi.org/10.1124/molpharm.123.000710>), thus in the basal condition the atypical receptor is constantly shuttling between the PM and inside the cell providing the mostly likely pool for receptors to be trapped on the surface.

Second, our group and others have previously shown that the inclusion of cycloheximide had no effect on the recovery of surface ACKR3 following chemokine stimulation within similar timeframes of our experiment (<https://doi.org/10.1038/onc.2010.212>, <https://doi.org/10.1371/journal.pone.0034192>). This suggests that new synthesis is unlikely to impact the accumulation observed during our experiment. An explicit comment to this has been added to the results:

“The accumulation is on a short enough time scale to suggest that the newly trapped receptors were previously constitutively internalized rather than de novo protein synthesis as ACKR3 shows little change in surface levels with hours of cycloheximide treatment (<https://doi.org/10.1038/onc.2010.212>, <https://doi.org/10.1371/journal.pone.0034192>).”

Finally, we repeated the experiment in Fig. 4B (PM bystander) in the presence of cycloheximide and saw no effect on the accumulation caused by VUN701, suggesting the increased surface BRET is not due to new proteins.

The lack of depletion from EE by VUN701 is therefore more likely due to changes in the trafficking of the receptor. We have made this point more directly in lines 234-236:

“The robust trapping of ACKR3 at the plasma membrane without depletion from early endosomes by VUN701 suggests that ACKR3 may be trafficked differently depending on receptor conformation and the neutral antagonist prevents a separate internalization mechanism apart from the early endosomal pathway.”

Introduction line 81: please add to the list of "handful" inhibitors ref (actual 83) doi: 10.1096/fj.202002465R, a real inverse agonist. This small molecule antagonist is being tested and is commercially available. The mechanism of action appears similar to VUN700 and VUN720 which should be added to the discussion as potential alternative to nanobody treatment.

We thank the reviewer for the remark. The compound ACT-1004-1239 is already referred to in reference 37 <https://doi.org/10.1021/acs.jmedchem.0c01588> on line 81. The suggested reference has been added to the list as well.

We have also added the following to the discussion:

“Alternatively, recently reported small molecule inverse agonists provide another avenue for targeting the constitutively active ACKR3 state (<https://doi.org/10.1101/2024.12.30.630720>, <https://doi.org/10.1096/fj.202002465R>).”

Reviewer #2 (Remarks to the Author):

The authors have carefully revised their manuscript. Additional data improved the study. A drop of bitterness is that no functional, in vivo data have been provided to address the consequences of keeping ACKR3 in its constitutive active conformation as asked for in the first round of revision. Instead of extending their observation on dampened MDA-231 breast cancer cell motility upon treatment with the inverse agonistic nanobodies, the authors have moved that data to the supplement and down-toned the potential functional importance of a constitutive active atypical receptor. As a “tumor model” HeLa cells have been used in the revised manuscript. If in vivo application of nanobodies may is not expedient, the small molecule inverse agonists (e.g. VUF16840 or CCX771) could be used instead. In essence the revised manuscript profoundly

describes the solid and careful in vitro characterization of two novel inverse agonistic nanobodies and provides new insights into the regulation of ACKR3 signaling using these nanobodies.

We are pleased that the reviewer finds the manuscript to solidly and carefully describe ACKR3 constitutive regulation as probed by inverse agonistic nanobodies. We consider an extensive in vivo study to be beyond the scope of this manuscript, however we plan to move in that direction in future studies. In this case, small inverse agonistic molecules could be taken along or used instead, as our results match closely the unpublished results using the small molecule inverse agonist VUF16480 (<https://doi.org/10.1101/2024.12.30.630720>) (CCX771 is a partial agonist <https://doi.org/10.1124/molpharm.121.000295>).

Minor points:

Lane 36: ..in inactive receptor conformation with decreased arrestin engagement..

The line has been changed to “These new tools promote an inactive receptor conformation which decreased arrestin engagement and inhibited constitutive internalization.”

Lane 96: Inhibition of receptor constitutive activity resulted in to slower cell motility. Fig 1F is not mentioned in the result section

We thank the reviewer for noting the omission and a call to Fig. 1F has been added to line 140.

Lane 164: Binding of the all nanobodies protected the extracellular face from deuteration,..

The extra ‘the’ has been removed.

Lane 238: What is the evidence for the statement that constitutive internalization of ACKR3 is independent of arrestins and GRKs?

We have shown this previously in <https://doi.org/10.1124/molpharm.123.000710>. This citation was included in this line and has been repeated to encompass the aforementioned statement.

Lane 254: reference 61 is a review article that does not show experimentally that ACKR3 contribute to cancer cell migration.

We agree that referring to a publication with experimental data would have been better. Therefore, we have added the following references to experimentally back up the statement: DOIs: [10.1016/j.neuron.2010.12.006](https://doi.org/10.1016/j.neuron.2010.12.006), [10.1242/dev.104224](https://doi.org/10.1242/dev.104224). We still included the review article for intrepid readers who want to find more.

Legend to Fig 1 has not been changed as promised in the rebuttal letter.

We apologise for the oversight, the change appears to have been lost between versions and editors. This has been corrected.

Reviewer #3 (Remarks to the Author):

The authors have addressed the large majority of comments, but a major concern remains.

There is a point that is still unclear to me, for which the authors did not properly comment:

In the HDX-MS experiment, how can the N-terminal peptide 27-33 be more deuterated after 30 sec labelling than after 30 min and 2 hours ? This phenomenon is only seen for the apo receptor. Even if seen throughout multiple replicates, this effect cannot be justified by any rational explanation. It rather suggests that the wrong peptide was used as a reference by the software. As exclusively this condition is used to claim that the nanobodies protect/ contact this region, I would not feel comfortable in having these data published.

We appreciate the reviewer bringing this aspect to our attention. This is a consequence of a commonly observed limitation of the technical setup. Due to time limitations at deuteration time-points below 2 minutes, the mixing syringes of the LEAP robot skip a cleaning step in non-deuterated buffer during sample preparation. This results in a lower overall back-exchange for the shorter time-points (max deuterated at 30 sec), which is only visible in non-structured, highly flexible regions. This technical limitation is routinely observed by us (for example: Grison et al., 2021, PNAS, doi: [10.1073/pnas.2108856118](https://doi.org/10.1073/pnas.2108856118)) and by others (Lumpkin and Komives, 2019, MCP, doi: [10.1074/mcp.TIR119.001731](https://doi.org/10.1074/mcp.TIR119.001731)). This observation does not refute our main conclusion, but rather on the contrary, consolidates them: seeing this only for the apo state and not for the Nb-bound state correlates with the fact that in the apo state the N-terminus is unstructured and becomes more structured in the presence of the Nb. Additionally, all peptides identified by the software were confirmed by manual inspection and the raw data for these datasets are available for review on PRIDE for further analysis.

Reviewer #4 (Remarks to the Author):

The authors have effectively addressed my comments and concerns. The resulting manuscript will be interesting and valuable to the research community.

One minor request is listed below:

*Information on the source and product numbers for the nanobodies used in this study (from QVQ) should be listed in the methods section.

The nanobodies will be made available following acceptance and publication of this manuscript and thus do not yet have product numbers. The designators for the nanobodies used by QVQ are included in the materials and methods: “VUN700 (product number Q125), VUN701 (product number Q123) and VUN702 (product number Q126).”